# Power pose effects on approach and avoidance decisions in response to social threat

**Hannah Metzler** [1,2,3¤]*, **Emma Vilarem**[1], **Adrian Petschen** [1,4], **Julie Grèzes**[1]*

**1** Cognitive and Computational Neuroscience Laboratory, (LNC2), INSERM Unit 960, Department of Cognitive Studies, Ecole Normale Supérieure, PSL University, Paris, France, **2** Sorbonne Universités, UPMC University Paris 06, Paris, France, **3** Institute for Globally Distributed Open Research and Education, **4** Institut de la Communication et de la Cognition, Université de Neuchâtel, Neuchâtel, Switzerland

¤ Current address: Complexity Science Hub Vienna, Vienna, Austria
* hannah.metzler@posteo.de (HM); julie.grezes@ens.psl.eu (JG)

**Data Availability Statement:** Data, code and material that support the findings and analyses of this study, are openly available at: https://osf.io/

## Abstract

Individuals' opportunities for action in threatening social contexts largely depend on their social power. While powerful individuals can afford to confront aggressors and dangers, powerless individuals need others' support and better avoid direct challenges. Here, we investigated if adopting expansive or contracted poses, which signal dominance and submission, impacts individuals' approach and avoidance decisions in response to social threat signals using a within-subject design. Overall, participants more often chose to avoid rather than to approach angry individuals, but showed no clear approach or avoidance preference for fearful individuals. Crucially, contracted poses considerably increased the tendency to avoid angry individuals, whereas expansive poses induced no substantial changes. This suggests that adopting power-related poses may impact action decisions in response to social threat signals. The present results emphasize the social function of power poses, but should be replicated before drawing strong conclusions.

## Introduction

Bodily expansiveness is closely linked to power-related behaviors in many social species, such as chimpanzees, rodents, birds, wolves, dogs, and crickets [1–5]. Large body size implies physical strength and often signals threat, whereas small bodies signal submission and vulnerability [6–9]. Humans use analogous non-verbal displays to express social power. Expansive body poses signal high power, dominance, prestige, and success [10–13], while contracted poses convey low power, submission, and defeat. Dominance and prestige are two distinct means of achieving social power [14]. While they are both related to body expansion, upright vs. slumped postures were suggested to be associated with prestige, whereas expansive vs. contracted power poses would be associated with dominance [13, 15]. Moreover, direct eye gaze, the absence of smiling behavior, and larger distances between hands and feet would more specifically communicate dominance [16].

q8s3w/ and https://github.com/hannahmetzler/poweraction.

**Funding:** J.G., H.M., E.M. as well as experimental and laboratory costs were supported by FRM Team DEQ20160334878 (https://www.frm.org), the Fondation ROGER DE SPOELBERCH (www.fondation-roger-de-spoelberch.ch), INSERM(www.inserm.fr), ENS (www.ens.psl.eu), the Agence Nationale pour la Recherche (www.anr.fr) under Grants ANR-20-CE28-0003, ANR-17-EURE-0017 and ANR-10-IDEX-0001-02. H.M. received a doctoral fellowship from the École des Neurosciences de Paris Ile-de-France (www.paris-neuroscience.fr) and the Région Ile-de-France (DIM Cerveau et Pensée, www.dimcerveaupensee.fr).

**Competing interests:** The authors have declared that no competing interests exist.

Building on embodiment theories, researchers have investigated the influence of body expansion on individuals' own feelings, moods, and behavior. Carney, Cuddy, and Yap [17] first tested whether adopting power poses for two minutes could significantly affect subjective feelings of power and control, risk-taking behavior, and levels of testosterone and cortisol. Although they found confirmatory evidence, later studies could not reproduce these results [18–26]. This sparked an intense debate about whether any of the later published power pose effects were reliable. Two recent meta-analyses provide evidence of a statistical difference between expansive and contracted poses for self-reported feelings and overt behavioral outcomes, but not for hormones, after correcting for publication bias (n = 96 reports [15], n = 48 reports [27], but see 28]). Both highlighted a lack of studies that allow definitive conclusions about whether expansive or contracted power poses drive the effects. The most reliable available evidence to date, from large pre-registered studies and registered reports, suggests that the power pose effect on self-reported feelings of power is real, but small (n = 1002 participants [29], n = 1071 participants [30]). Still, it seems likely that the effect arises partly from demand effects [15, 30, 31], and that publication bias remains an issue in the field [15, 32].

The present study addresses some of the common issues in power posing studies: It assesses changes from a baseline to power-related poses [15, 27] and uses behavioral outcome measures and a plausible cover story rather than self-reports and explicit instructions that could increase demand effects [30, 31]. Crucially, our study focuses on lower-level processes of social interactions, specifically the ability to perceive and respond to others' social signals. With one exception [33], earlier work using behavioral outcomes has so far neglected such low-level social behaviors, focusing either on cognitively complex (and sometimes only reported) behaviors [20, 34] or non-social behaviors [35, 36].

Recent research has shown that adopting power poses can influence the perceptual salience of emotional facial displays [37]. In everyday life, emotional facial expressions not only inform others about the affective states and potential behavioral intentions of the emitter [38] but also convey action demands to the perceiver [39]. Accordingly, perceiving emotional expressions has a direct influence on the observer's behavior [40], prompting motivational orientations that prepare the organism for appropriate responses. The present study investigated whether adopting power poses would also influence action decisions in the presence of emotional facial displays.

To do so, we used a novel approach-avoidance paradigm developed by Vilarem et al. [41], which involves spontaneous decision-making between two competing targets for action in a socio-emotional context. In depicted everyday scene of a waiting room with four chairs, two individuals sit on the two middle ones, one of them expressing an emotion. Participants were requested to choose which seat they would like to occupy in the scene. By choosing the outer chair further away or next to the emotional individual, the task allows participants to increase (avoid) or decrease (approach) the distance to the emotional individual. It differs from previous approach-avoidance paradigms [42], during which participants are instructed to perform one movement (e.g. push or pull), which either results in approaching or avoiding valenced stimuli. In contrast to such forced-choice tasks here, participants freely choose between two action possibilities in the depicted scene, without the confounding effect of instructions, arbitrary movements, or response labels (see S1 Appendix for a detailed comparison to previous approach-avoidance tasks). To assess postural feedback effects, we ran two consecutive sessions of this task: the first session, without poses, served as a within-subject baseline, whereas participants adopted either an expansive or a contracted pose before each task block in the second session. Given that social power is fundamentally about defending access to resources needed for survival against others [43, 44], we focused on threat-related facial expressions of anger and fear. Encountering conspecifics that express threat-related facial expressions represents a potential menace to an individual's resources and/or survival. The individual's power

will crucially determine how they can respond to social threats [44, 45], that is their action opportunities. While anger and fear are both of negative valence, they convey different social meanings [46, 47]. Facial expressions of anger, by enhancing cues of strength and dominance [48], are signals of aggressive intent [49], a clear threat to the observer [46], which in most contexts leads to avoidance. Fearful displays, in contrast, signal both the presence of a potential danger [50] and a need for affiliation [51, 52], and are thus more ambiguous in terms of avoidance and approach behaviors. Contrasting these two emotional expressions thus allowed determining whether power poses differently influence decisions based on clear versus ambiguous and subtler social cues.

Previous studies using this task [41, 53, 54] observed that participants tend to avoid angry individuals more often than fearful ones, without having been informed about the emotions, and most often, without being able to explicitly report which emotions were presented during the task. Both angry and fearful faces elicited quicker reactions than neutral faces, hinting that the increased avoidance of angry compared to fearful faces cannot be explained by more efficient processing. Mennella et al. [54] and Grèzes et al. [53] further revealed that angry and to a lesser extent fearful displays, increased the expected value of the action leading to avoidance. In other words, sitting far from a threatening individual was a highly motivational (desirable) outcome per se, even in the context of a laboratory task.

Research further suggests that action possibilities are constrained by the individual's action capabilities [55], which can be rapidly recalibrated as a function of external [e.g., 56] or internal factors [57, 58]. We, therefore, hypothesized that power poses could impact the computation of available action possibilities and associated expected value according to the level of power they embody, without influencing the ability to detect anger or fear.

Powerful individuals can afford to approach aggressive others and confront them, whereas powerless individuals may be better off avoiding conflicts and seeking social support as a means of protection. Greater physical and social resources [44] enable powerful individuals to better cope with social threats, such as an angry opponent. Moreover, power increases approach motivation in general [44, 45] and decreases vigilance toward threat [59, 60]. Manipulations of power induce opposite approach and avoidance tendencies in response to sustained direct eye gaze, another social threat and dominance signal [61]. Therefore, if expansive poses embody high power, they should decrease the avoidance of angry individuals, in contrast to contracted poses, which should increase avoidance.

Given the ambiguity of fearful displays, which signal both the presence of danger and the need for affiliation, our predictions for fear were more speculative. Knowing that the lack of power increases affiliation motivation [62] and subtle cues of low compared to high social status increase pro-social behavior [63], contracted compared to expansive poses may increase approach toward fearful individuals, who signal a need for help and represent potential allies in the defense against a threat.

## Methods

Data, code, and materials to reproduce the below procedures and analyses are available at https://osf.io/q8s3w/ and https://github.com/hannahmetzler/poweraction.

### Participants and power analysis

A total of 88 healthy, right-handed, fluent French-speaking men between 18 and 35 years old were recruited via a mailing list and online student job platforms. All participants had a normal or corrected vision, were not currently under medical treatment, and did not suffer from ocular pathologies or ocular fatigue in front of a screen. The experimental protocol, approved

by INSERM and the local research ethics committee (Comité de protection des personnes Ile de France III—Project CO7-28, N° Eudract: 207-A01125-48), was carried out in accordance with the Declaration of Helsinki. Participants provided informed written consent and were paid for their participation.

In the context of the debate around the replicability of previous power-posing studies (this study began in 2016), a major priority was to achieve high power within our feasibility constraints. Given gender differences in a study of ours with the same facial stimuli [64, Chapter 4] we decided to maximize the sample size per group by including only male participants. A recent meta-analysis suggests that power-pose effects do not differ between genders [15]. We calculated the required sample size to detect a significant pose by emotion interaction on the change in proportion of away choices from the first to the second session at an alpha of .05 and .80 power. We based the calculation on the smaller out of two previously observed effects of emotion in the same task, i.e. partial eta-squared of $\eta^2_p = .30$ compared to $\eta^2_p = .36$ [41, 54]. Entering this effect size (as f = 0.65) in a one-way repeated measures ANOVA in G*Power [65] yielded a minimal sample size of n = 22 to detect the emotion effect, which corresponds to 22 per group to detect the interaction with pose if poses completely suppressed the effect in one group [66]. Yet, we expected the emotion effect to be larger in the contracted pose and smaller in the expansive pose, with the actual extent of this change being unknown. Therefore, we doubled this number, aiming at 44 participants per pose group. The final sample consisted of 79 participants (see section on data cleaning) with a mean age of 22.70 ± 3.64 years, of which 40 and 39 had been randomly assigned to the expansive and contracted condition, respectively. See the S1 Appendix for a sensitivity analysis to determine the achieved power with this final sample size.

## Stimuli and task

Stimuli [41] were presented using the Psychophysics Toolbox 3 [67, 68] in MATLAB version 2014b [69]. They depicted a waiting room with four chairs, with two individuals seated on the two chairs in the middle. One of the two depicted individuals always displayed a neutral facial expression while the other displayed, in one third of the trials, either a neutral, angry, or fearful expression. The stimuli included facial expressions from 10 different pairs of individuals from the Radboud Faces Database [70], which were matched for gender as well as for perceived threat and trustworthiness [41]. Expressions of anger and fear varied along 4 levels of intensity (morphs between the neutral and the full emotional expression), which were equalized for perceived emotion intensity [71]. Which individual in each pair expressed an emotion, and on which side (left/right) in the stimulus image it was sitting, was fully counterbalanced. Each individual in a pair once displayed each of the 2 emotions at 4 different intensity levels and displayed the same neutral expression 4 times. Each individual was further once displayed on each side in the image (left/right) with each of these expressions. With 10 pairs of two individuals, this resulted in a total of 480 trials per session: 10 pairs x 2 individuals x 2 sides x (2 emotions x 4 intensities + 4x1 neutral expression).

Participants' task was to choose on which of the two outer chairs they wanted to sit, by moving the mouse cursor from the middle of the screen to one of the two outer chairs. Throughout the trial, they had to keep their eyes on the fixation cross displayed between the two faces. They were told to choose the chair as if making the choice in a real situation, and that there was no good or bad choice, as long as they released the cursor within one of the chair areas. Participants were asked to maximize the number of valid trials, i.e. to land within one of the chair areas within 1400ms. Their accuracy score (percentage of valid trials) was displayed at the end of each block. The timing of a trial is illustrated in Fig 1.

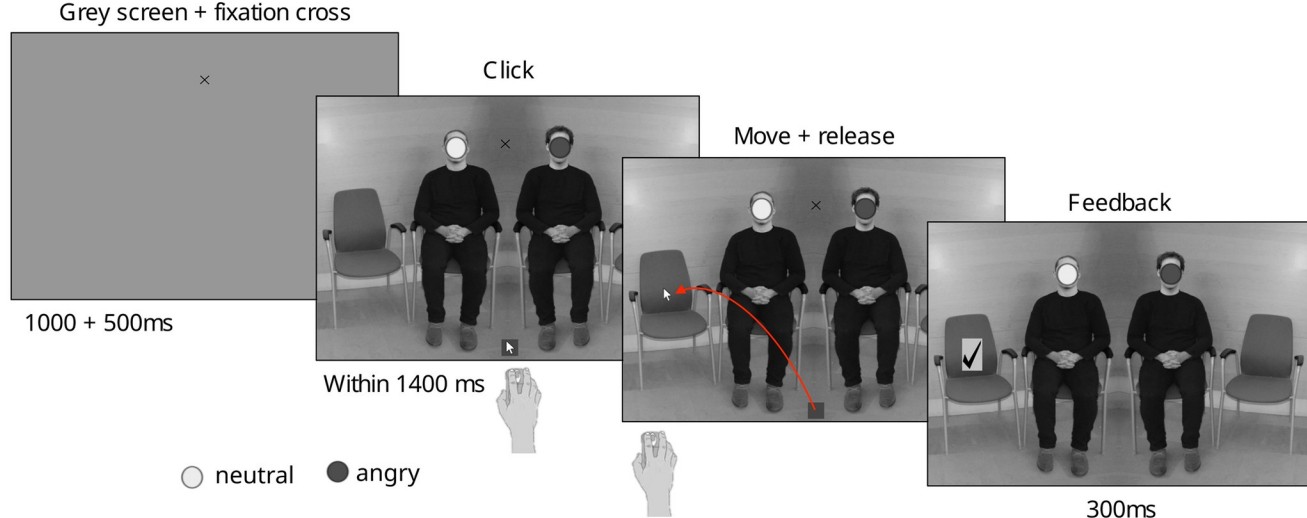

**Fig 1. Stimuli and task.** Time course of a trial where participants were asked to indicate where they would like to sit. The faces were masked due to copyright issues. Stimuli depicted in the experiment included faces from the Radboud Faces Database. For a figure without masks, see Fig 1 in [41] or in Chapter 7 in [64].

In total, there were 10 blocks of 96 trials with a mean duration of 6.5 ± 1.95 SD minutes. Participants achieved mean accuracy scores of 91.3% ± 5.37 SD. Each trial started with a gray screen of 1000ms, onto which a fixation cross was then superimposed for 500ms, followed by the scene with the mouse cursor at the bottom center. Throughout the trial, participants were to focus on the fixation cross displayed between the faces. Upon presentation of the waiting room, they had to click, hold and move the mouse cursor from the gray square to the chair of their choice, and release it within the chair area. In valid trials, i.e., when releasing the cursor within the chair area within 1400ms after scene onset, a tick appeared at the release location for 300ms before the next trial started. In case of releases outside of the chair area, the next trial started immediately, or after 1400ms if no response was made.

Participants were seated so that the distance from the eyes to the screen was 60 cm. Like this, the eccentricity to the central fixation cross was 4.5 degrees for the center of the faces, and 8 degrees for the center of the chairs. Participants were instructed to fixate on the fixation cross throughout the trial and were not informed that the individuals depicted in the trial would express any emotions. During a brief training session with only neutral facial expressions, participants were asked to land the cursor on a blue rectangle above each chair. The training was repeated if necessary until they landed on the chair correctly in at least 60% of trials.

## Procedure

To ensure that relevant trait or state differences between the two pose groups did not contribute to observed group differences, participants completed questionnaires before the testing day, and just after signing consent forms (see the S1 Appendix for further details and results). Participants performed the first 5 task blocks without pose (session 1), and then stood in either an expansive or contracted pose (see Fig 2) for 2 minutes before each of the next 5 task blocks (session 2).

To justify this experimental design, the male experimenter told participants that they were participating in two separate studies, the first investigating spontaneous action choices and the

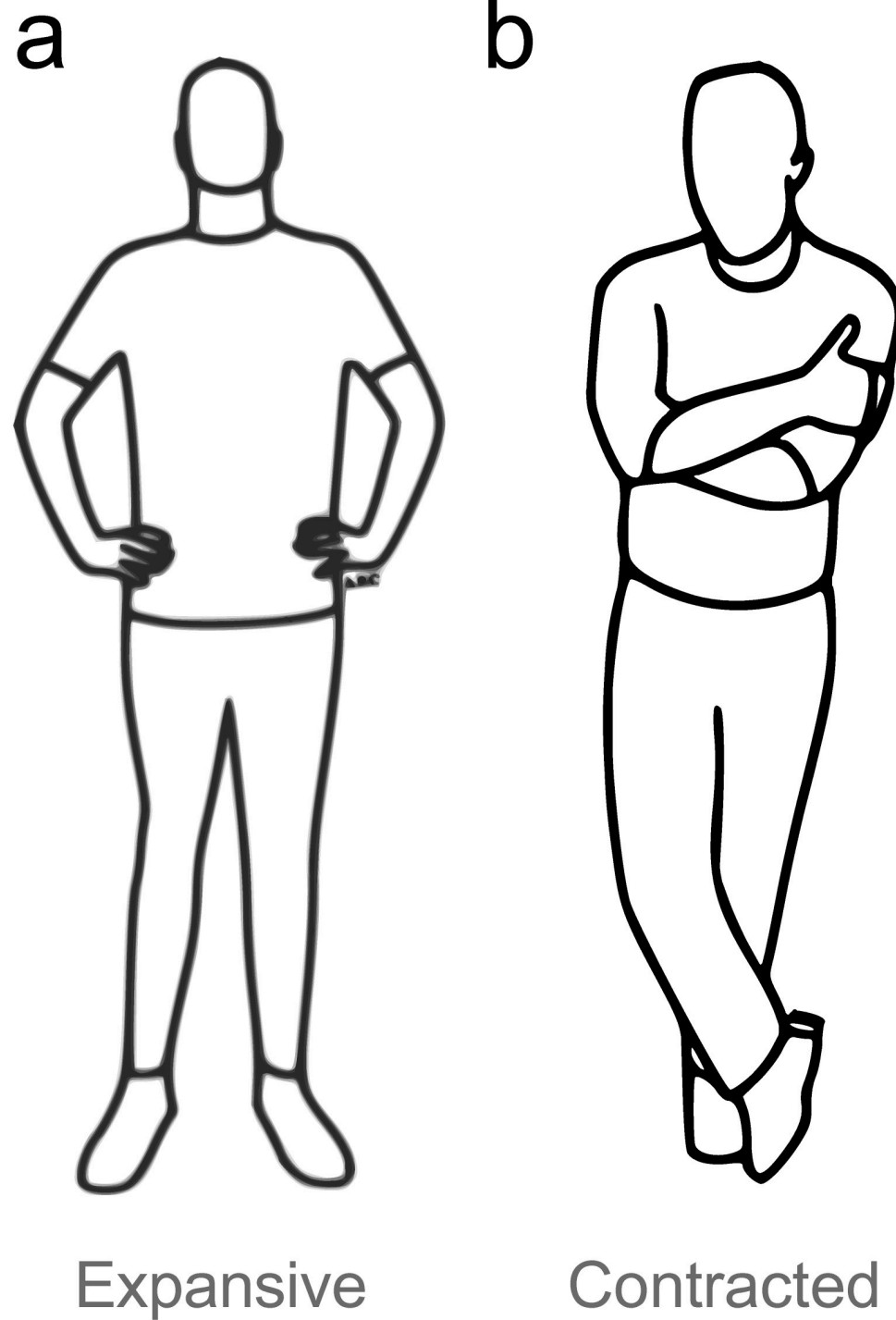

**Fig 2. Poses adopted by each of the experimental groups.** Participants adopted no pose in session 1 (baseline). In session 2, they adopted one of the depicted poses for 2 minutes before each of the 5 task blocks. (a) Expansive pose. (b) Contracted pose. Legend: Images created by Antoine Balouka-Chadwick.

second posing effects on heart rate. The cover story further mentioned several advantages of combining the two studies, such as breaks from screen time in the second half of the experiment and short posturing intervals to avoid discomfort. After session 1, the experimenter attached electrodes to participants' wrists and demonstratively turned on the heart rate acquisition system, although no data were recorded. To minimize possible experimenter biases, participants were randomly assigned to a pose condition just before pose instructions were provided (see S1 Appendix for more details on instructions).

Careful debriefing at the end of the experiment suggested that only 12 participants had noticed an influence of facial expressions on their decisions, none could specifically identify anger and fear as the two displayed emotions, and only 2 participants suspected a link between the pose and the task. Excluding these 2 participants did not change the results. Such low familiarity with power posing may be due to the fact that all sessions were conducted between November 2016 and March 2017, when media coverage of power poses in France was low. For more details, see the S1 Appendix.

### Data analysis

After exporting data from Matlab, all analyses were conducted in R version 3.4.4 [72] using packages from the Tidyverse, ez, and psych [73–75], in addition to those specifically mentioned below. We recorded the chosen chair (coded as moving away from or toward the emotional individual), and the movement kinematics initiation time (time from scene onset to mouse click), and movement duration (time from mouse click to release within the chair area).

### Data cleaning

Only valid trials were included in the analysis (landing on top of a chair area and under 1400ms). Additionally, we removed two types of trials that reflected a lack of compliance with task instructions: trials in which participants anticipated the scene onset due to the fixed duration of the inter-trial interval and clicked too early (trials with initiation times below the 5th percentile, i.e. below 95ms) and trials in which participants clicked after the beginning of the movement (trials for movement durations below the 5th percentile, i.e. shorter than 184ms). To ensure a sufficient number of trials per participant, we required at least 50% of valid trials per session above the respective 5th percentile threshold for both initiation time and movement duration, which entailed the exclusion of 6 participants. Excluding these participants, of which 3 were in the expansive and contracted condition, respectively, did not affect the significance of effects in the model of choice, initiation and movement time. Due to a technical error, data from 2 further participants were lost. Another participant was excluded because he did not achieve 60% accuracy after repeating the training three times (see task description). The mean trial number per emotion condition (anger, fear, neutral) in each session for all remaining 79 participants was 132.6 ±17.22 with a range of 65–158 trials. Finally, initiation time and movement duration were log-transformed to account for the obvious right skew typically observed in reaction time distributions. All 95% confidence intervals in figures and tables are within-subject intervals, calculated separately for each group according to the Cousineau-Morey method [76] in order to highlight pre-post pose changes.

### Proportion of choice

Analyses of choice (away vs. toward), and movement kinematics were performed using linear mixed-effects models (LMEM) as implemented in the lme4 package [77] to account for repeated measures and the unbalanced design (unequal number of participants per pose and

valid trials per condition), and to consider random variation between participants as well as stimuli. For choice, we analyzed the effect of emotion (anger vs. fear, within-subject), session (session 1 baseline vs. session 2 with pose, within-subject), and pose (expansive vs. contracted, between-subject). We opted not to examine the effect of intensity as we had no specific hypothesis regarding intensity and pose, and models without intensity already involved interactions of three and four predictors for choice and kinematics, respectively.

We used deviation coding (-0.5, 0.5) for emotion, with fear as the baseline category. In contrast, session and pose were treatment coded (0,1) with session 1 and expansive as default baselines. With this combination of contrast coding, the model intercept corresponds to the mean of the dependent variable across emotion in the baseline category, i.e. in the expansive group in session 1. The estimate for emotion represents a main effect, whereas estimates for session and pose correspond to simple effects (i.e., they reflect the change from session 1 to 2, and the difference between pose groups). To obtain effect estimates for main effects and two-fold interactions in the contracted group and session 2, we re-leveled pose and session and ran each model once with each pose and session as the baseline category.

We applied a generalized LMEM with binomial family and logit function on only angry and fearful trials (choice away or toward the emotional actor is not meaningful in neutral trials), with main effects and all interactions of emotion, session, and pose as fixed effects, and a random intercept for participants and stimuli (see S1 Appendix for more details on the random effects structure). To interpret our main result with regard to the statistical power, we further report an ANOVA of the proportion of choice with session, emotion, and pose in the S1 Appendix. Details on the analysis and the results regarding movement kinematics are reported in the S1 Appendix.

## Results

### Proportion of choice

A likelihood ratio test comparing the generalized LMEM *choice ~ (1|subject) +(1|stimulus pair) + emotion x session x pose* to the null model with only random intercepts, indicated a good fit to the data ($\chi^2(7)$ = 35.789, p < .001, deviance (-2LL) reduced from 57854 to 57818). Fig 3a illustrates that there was a higher probability of avoiding anger compared to fear in both sessions and groups. However, the predicted three-fold interaction suggested that this effect of emotion on action choices (more avoidance for anger than fear) changed between sessions as a function of adopted pose (OR = 1.19, 95% CI[1.02, 1.38], z = 2.18, p = .029). Specifically, the difference in avoidance of anger and fear was larger in the second than the first session in the contracted compared to the expansive pose (see S1 Appendix for results and effect sizes in the form of ANOVAs and Cohen's d for central comparisons).

Furthermore, Fig 3 illustrates that this change was predominantly driven by responses to anger, than responses to fear. In participants who adopted contracted poses, the probability of selecting the chair farther away from the angry individual increased by 2.71% on average, while it only decreased by 1.23% for fear. In participants who adopted expansive poses, there were no substantial changes for either anger (0.28%) or fear (-0.23%, see Table 1 for means and confidence intervals).

To evaluate the significance of these changes from sessions 1 to 2 in each pose group, we compared the emotion-by-session interaction when the expansive and contracted group was set as the baseline category of the treatment contrast, respectively. This interaction was only significant in the contracted group (OR = 1.14, 95% CI [1.02, 1.27, z = 2.30, p = .022, expansive group: OR = 0.96, CI [0.86, 1.07], z = -0.78, p = .435) suggesting that only contracted poses induced a significant change. Indeed, the main effect of emotion was not significant in the

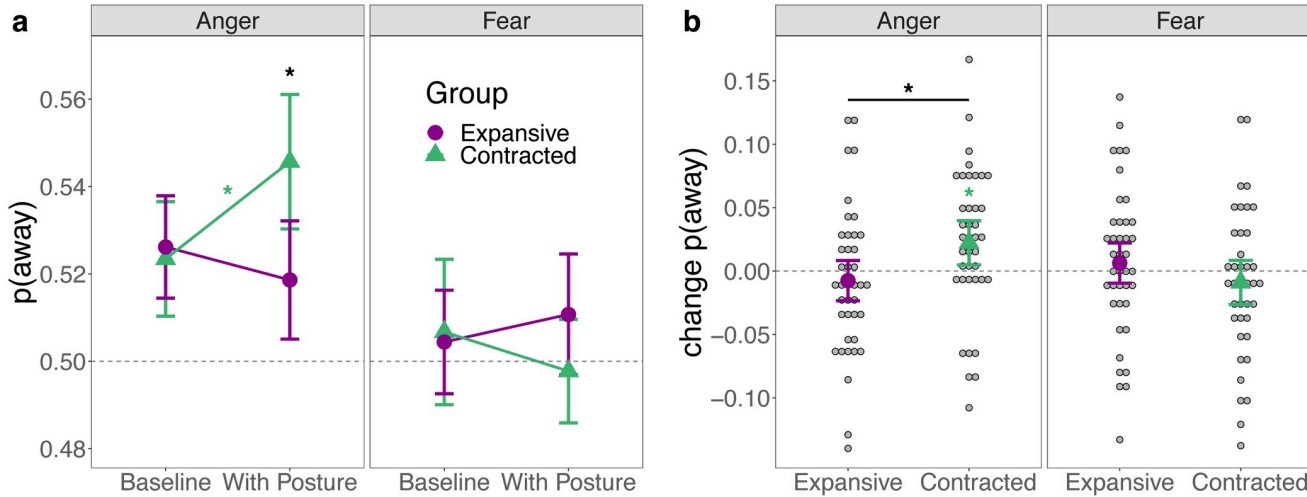

**Fig 3. Means and within-subject confidence intervals for proportion of choice.** p(away): Proportion of trials in which participants moved away as opposed to toward the emotional individual. (a) Proportion in each pose and session (session 1: baseline, session 2: with pose). Contracted poses significantly increased avoidance of anger, whereas expansive poses induced no significant change. In response to fear, there were no significant changes. (b) Change in proportion of choice from session 1 to session 2 in each pose. Only the change in response to anger in the contracted pose was significant. A confidence interval not overlapping with 0 indicates a significant change. * = p < .05 in within-subject (color) or between-subject (black) t-tests.

contracted group in session 1, but highly significant when participants adopted the contracted pose during session 2. In the expansive group, the effect of emotion was significant in session 1, but not anymore in session 2 (see Table A in S1 Appendix for details). Similar odds ratios for the emotion effect in both groups in session 1 (OR = 1.07 and OR = 1.09) indicate that the emotion effect was comparably large at baseline in both groups.

Comparing the emotion by pose interaction with sessions 1 and 2 as baseline of the contrast, respectively, confirmed that the effect of emotion did not differ between groups in session 1 (OR = 0.99, 95% CI [0.88, 1.10], z = -0.27, p = .789), but in session 2 (OR = 1.17, CI [1.05, 1.30], z = 2.81, p = .005). This suggests that the difference in session 2 was not due to baseline group differences in the proportion of away vs. toward choices. Finally, neither the simple effects of session and pose nor the session-by-pose interaction was significant regardless of which pose or session was set as the baseline for the respective contrasts. This indicates that

**Table 1. Descriptive statistics and confidence intervals for proportion of away choices.**

| Session | Group | Emotion | n | Mean | SD | 95% CI |
|---|---|---|---|---|---|---|
| 1—Baseline | Expansive | Anger | 40 | 52.62 | 4.1 | 51.44–53.79 |
| 1—Baseline | Expansive | Fear | 40 | 50.44 | 3.86 | 49.26–51.63 |
| 1—Baseline | Contracted | Anger | 39 | 51.86 | 5.13 | 50.51–53.21 |
| 1—Baseline | Contracted | Fear | 39 | 51.07 | 4.12 | 49.69–52.46 |
| 2—Pose | Expansive | Anger | 40 | 52.34 | 5.54 | 51.03–53.65 |
| 2—Pose | Expansive | Fear | 40 | 50.67 | 4.33 | 49.01–52.33 |
| 2—Pose | Contracted | Anger | 39 | 54.57 | 6.47 | 53.03–56.11 |
| 2—Pose | Contracted | Fear | 39 | 49.77 | 3.56 | 48.59–50.96 |

The percentage of trials where participants chose the chair away from the angry or fearful individual. Within-subject, 95% confidence intervals were calculated separately for each group according to the Cousineau-Morey method.

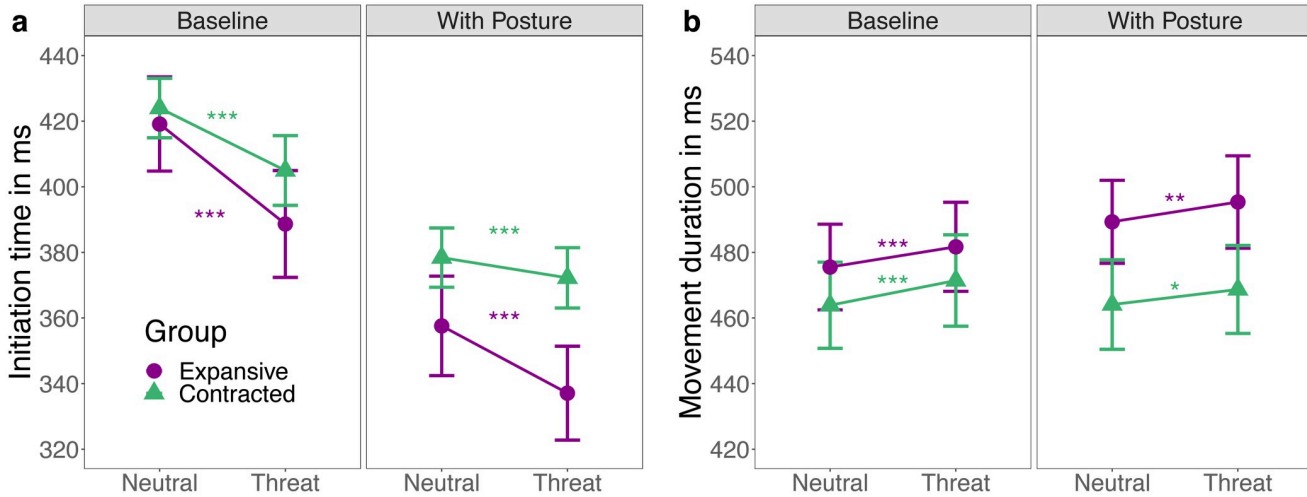

**Fig 4. Initiation time and movement duration.** Means and within-subject confidence intervals for (a) initiation time (from presentation of scene to click) and (b) movement duration (from click until release on top of a chair) in ms per pose and session (session 1: baseline without pose, session 2: with pose). (a) In both sessions and groups, initiation time was quicker for trials in which one actor expressed threat (fear or anger), compared to trials with two neutral actors. Initiation times were further quicker in the second session, with a stronger acceleration in the expansive group. (b) In both sessions and groups, movement duration was significantly longer for trials in which one actor expressed threat (fear or anger) compared to trials with two neutral actors. In the expansive group, movement duration was further slower in the second compared to the baseline session. For neither (a) nor (b), any between-pose comparisons were significant. Note that asterisks indicate significance of the threat effect in model comparisons at * p < .05; ** p < .01 and **p < .001. See Table G in S1 Appendix for all other effects.

there were no general differences in the proportion of choice between sessions or groups (e.g. more avoidance than approach in general; see Table A in S1 Appendix for all model estimates and significance tests).

In summary, approach and avoidance decisions in response to angry and fearful individuals did not differ between the two groups before participants adopted a pose. All participants avoided angry more than fearful individuals. After adopting a contracted pose, the probability to avoid anger compared to fear became larger, whereas expansive poses induced no significant changes in action choices.

**Initiation time and movement duration.** The LMEMs for movement kinematics revealed no significant difference between angry and fearful trials, and no interaction of emotion with pose (see S1 Appendix). Grouping anger and fear together to compare threat with neutral trials revealed quicker initiation time and slower movement duration in response to threat (see Fig 4). This suggests that threat-related facial expressions significantly reduced the time needed to initiate an action. More details and other results for movement kinematics are reported in the S1 Appendix.

## Discussion

The current study assessed whether adopting expansive or contracted poses, which function as social signals of power and dominance, impacts individuals' approach and avoidance decisions in response to threat signals emitted by other individuals. More specifically, we investigated free action choices in the presence of individuals who displayed task-irrelevant angry or fearful facial expressions. Uninformed about the emotional expressions, participants chose where to sit in a realistic scene by moving the mouse cursor onto a chair, thereby either approaching or avoiding the emotional individual. Following a first baseline session without poses, participants repeatedly adopted either an expansive or contracted pose between task blocks during a

second session. In contrast to the outcome measures used in most previous power pose studies, action decisions in response to threat signals constitute actual rather than reported behaviors related to the social function of these poses. In addition, such elementary behaviors in response to task-irrelevant social stimuli may be less susceptible to experimental demand effects than explicit self-reports and require no high-level cognitive processes.

Across groups, the presence of angry individuals, who signal a direct threat towards the observer [46, 48], favored the selection of avoidance actions. This replicates findings from experiments that used the same task [41, 53, 54]. Mennella et al. [54] further demonstrated that as sitting far from a threatening individual is a highly desirable outcome per se, the presence of such individual increases, before choice, the value of the action leading to their avoidance. In addition, our results suggest that power poses modulated action decisions in response to task-irrelevant angry displays in the predicted directions. Specifically, expansive poses decreased avoidance ($d_z$ = -0.13, non-significant change), while contracted poses significantly enhanced individuals' choices to avoid angry individuals ($d_z$ = 0.38, between-pose effect of d = 0.51, see S1 Appendix). The present results could indicate that contracted poses increase the value of the action leading to the most desirable outcome (threat avoidance), leading individuals to select actions that allow avoiding aggressive conspecifics. Similarly, by manipulating power using mindset priming via a writing task, Weick and colleagues [61] showed that the threat signal of sustained direct gaze elicited spontaneous avoidance tendencies in individuals with low power.

Expansive poses did not significantly decrease avoidance of angry individuals, in contrast to our predictions. While our result appears to be consistent with a recent meta-analysis suggesting that only contracted poses may change behavior in comparison to neutral poses [27], the issues raised about this meta-analysis [28], and the small number of studies including neutral conditions prevent any definitive conclusion at this point. One possible explanation for the absence of expansive pose effects relates to Guinote's [45] updated account of the approach motivation theory of power [44], according to which power activates people to pursue their goals and desires, and thus activates the approach system specifically in association with goal seeking. Therefore, expansive poses may not have increased approach behavior because there is no goal to achieve by approaching an angry individual in the context of a waiting room. When the choice of avoiding an angry person is available and comes without any disadvantage, confronting this person may simply not be adaptive.

In the presence of fearful individuals, who simultaneously signal the presence of a potential danger and a need for affiliation [78], participants' choices suggested neither an approach nor an avoidance preference. The changes induced by adopting expansive compared to contracted poses on approach and avoidance of fearful individuals were small and non-significant. Nonetheless, they occurred in opposite directions in the two groups: avoidance decreased in the contracted ($d_z$ = -0.15) and increased in the expansive poses group ($d_z$ = 0.11) resulting in a small between-pose effect (d = -0.26). The direction of this change, in case it reflected a true reproducible effect, would hint that participants were more sensitive to the affiliative than the threat aspect of fearful displays after adopting contracted compared to expansive poses, in line with research suggesting an increase in affiliation motivation in states of low power [62, 63]. Although we had reasonable statistical power to detect the observed difference between angry and fearful displays (75% for the interaction of pose and emotion), larger samples would be needed to test whether this small change in response to fearful displays (main effect of pose) is reliable.

As expected, the poses' observed influence on approach and avoidance decisions did not coincide with differences in the processing of threat-related displays. Consistent with previous studies [41, 53, 54], both pose groups were quicker to initiate their actions and took longer to

reach the chairs when responding to both threat-related displays compared to neutral ones [41, 53, 54]. These general threat effects suggest that anger and fear displays were processed to a similar extent. Therefore, the stronger tendency to avoid angry compared to fearful displays, which was reinforced after a contracted pose, cannot be explained by improved or quicker perceptual processing of anger compared to fearful displays. Instead, anger and fearful displays may differently shape action selection according to their social function [41, 47, 79]. Hence, the presence of task-irrelevant angry displays more strongly favored avoidance decisions than fearful displays because anger communicates an unambiguous aggressive intent and is perceived as a direct threat to the observer [48]. Likewise, adopting a contracted pose may enhance avoidance decisions to angry displays because the communicated threat might be perceived as more challenging when the observer is in a submissive position.

Consistent with research showing that individuals experiencing high power initiate goal-directed actions more readily [43, 80], the expansive group showed a larger threat effect for initiation time than the contracted group. High compared to low dominance individuals have been shown to make faster decisions in competitive settings [81–83], which may generalize across a variety of decision-making tasks in men [84]. Da Cruz et al. [84] suggested that prompt responses in decision-making situations without a loss of accuracy could be at the core of establishing social rank.

One possible interpretation of the power pose effects observed in the present study relates to action possibilities. The perceived available action opportunities and their assigned expected value are constrained by an individual's action capabilities [55] that are rapidly recalibrated as a function of external [e.g. 56] or internal factors (e.g., fatigue [57, 58]). When facing social threats, power poses could impact the computation of available action possibilities and associated expected value according to the level of power they embody. Appraisal theories also assume that appraisals depend on the person-environment relationship, which can change over time and circumstances [85–87]. Within this framework, power poses could change the appraisals of the motivational relevance of others' emotional expressions and an individual's perceived ability to react to the situation. A contracted pose may make it more difficult to confront a threatening other and therefore lead to increased avoidance. This explanation is consistent with the approach-motivation theory of power and related evidence, which implies that high-power individuals have more freedom to express their personality and emotions, whereas individuals with low power need to adapt their behavior to situational circumstances and suppress their emotions [44, 88, 89]. This could explain why individuals in a powerless state tend to avoid intense emotional and threatening confrontations. Other possible interpretations of increased avoidance of angry individuals could be related to a change in mood or effort while holding the contracted pose. Future studies could try to differentiate between these different possible mechanisms.

By demonstrating pose effects on elementary social behavior with reasonable statistical power, the present study makes a valuable contribution to the literature. Most importantly, it is the first larger study investigating elementary social behavior, with the only earlier study having a very small sample (n = 22 [33]). Although power poses are primarily a social signal of dominance in many animal species, most previous studies have focused on non-social behavior or high-level cognitive processes, such as risky gambling, abstract thinking, or sales negotiations [20, 90]. The results of the present study suggest that investigating elementary social behavior could be a promising direction for future research.

Additionally, the present study attempted to overcome some of the methodological limitations of previous studies on postural feedback effects. Past experiments that yielded significant effects tested small samples, whereas studies with large samples and pre-registered analysis plans mostly reported null-effects [20, 31]. Moreover, very few power posing studies have so

far used designs that allow attributing the change to the expansive or contracted poses by including a neutral condition. Within-subject studies observed either a change only after expansive poses [91, 92] or no significant effect [33]. Similarly, only a few studies included a control group, and observed a significant difference only for the contracted pose [36] or no significant effects [21, 26]. Thanks to the within-subject design of the current study, we can be more confident that the observed pose effects did not originate from baseline between-group differences in approach and avoidance tendencies.

Despite these strengths, this study is not without limitations. First, within-subject designs were suggested to increase the risk of demand effects [15]. In the past, the majority of studies on postural feedback effects used explicit self-reports about behaviors [93–96], which appear to be susceptible to demand effects. Indeed, the pose effect on explicit feelings of power was found to be larger in subjects familiar with "power posing", in a meta-analysis of six pre-registered and highly-powered studies [30], as well as in a similarly large pre-registered study [29]. In the present study, however, only 2 participants suspected a link between the pose manipulation and the approach-avoidance task even after extensive debriefing (see S1 Appendix), indicating that the influence of poses on action decisions in response to task-irrelevant stimuli may reduce such demand effects. Second, we only included male participants in the present study. This decision was due to the study being part of a larger project on power poses [64], which began before it was clear that effects on cortisol and testosterone did not replicate. We initially suspected hormonal mechanisms to underlie the poses' effects on social behavior, and biological differences in the production of testosterone and the menstrual cycle require separate hormonal analyses by gender [97]. Therefore, most studies in the project focused only on men, whereas those that included women observed gender effects in face perception [64, Chapter 4]. Wanting to achieve robust results and maximize sample size per group in the context of the replication debate, within our feasibility constraints, we therefore included only men. As Körner et al.'s [15] meta-analysis revealed that power pose effects do not depend on gender, our findings may generalize to women.

Finally, it is important to transparently report on the level of evidence this study provides for power pose effects on low-level social behavior. This requires mentioning its exploratory nature, and the debate about replicability especially for early research in the field (up to 2017). We believe it provides clear preliminary evidence due to its many strength (robust experimental and statistical methods, a baseline measure, open data, code, and materials, and strong theoretical foundation on dominance poses as a social behavior relevant in threatening contexts). Yet, we are convinced that the medium effect sizes, given our sample size, would require replication before strong conclusions can be drawn.

## Conclusions

This study provides evidence that briefly adopting power-related poses could impact individuals' approach and avoidance decisions in response to social threat signals. The observed influence on action decisions in response to angry, but not fearful individuals, was not explained by differences in processing between the two emotional expressions, but reflected a difference in their social function. Hence, postural feedback, by modulating the motivational relevance of certain threat signals to the perceiver, may not only affect visual attention [37] but also the decisions taken in response to such threat signals. Within the context of the controversy around the replicability of previously published pose effects [26, 31], the present results need to be replicated before any strong conclusions can be drawn. As elementary behaviors that are conceptually linked to the social meaning of expansive and contracted poses, approach and

avoidance decisions in response to threat signals may be among the most promising candidates for future replication efforts.

## Supporting information

**S1 Appendix. Supplementary information on methods and results.** Regarding the methods section, the S1 Appendix contains detailed information on how our task compares to other approach-avoidance tasks, the cover story, pose instructions, debriefing, self-report questionnaires and data analysis. In the results sections, we report self-report questionnaire values for both experimental groups, result tables for all generalized linear mixed effects models, results in terms of ANOVA and t-tests for proportion of choice, and all results for movement kinematics. These include descriptive statistics for movement kinematics, linear mixed effect model and ANOVA results for both reaction time and movement duration.
(PDF)

**S1 File.**
(PDF)

## Acknowledgments

We thank Rocco Menella and Julia Maria Carbajal for advice on analysing and plotting data with linear mixed effect models.

## Author Contributions

**Conceptualization:** Hannah Metzler, Julie Grèzes.

**Formal analysis:** Hannah Metzler.

**Funding acquisition:** Hannah Metzler, Julie Grèzes.

**Investigation:** Hannah Metzler, Adrian Petschen.

**Methodology:** Hannah Metzler, Emma Vilarem, Julie Grèzes.

**Project administration:** Hannah Metzler, Julie Grèzes.

**Resources:** Hannah Metzler, Emma Vilarem.

**Software:** Hannah Metzler, Emma Vilarem.

**Supervision:** Hannah Metzler, Julie Grèzes.

**Visualization:** Hannah Metzler.

**Writing – original draft:** Hannah Metzler.

**Writing – review & editing:** Hannah Metzler, Adrian Petschen, Julie Grèzes.

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
