## [Decision Letter · Decision Letter 0]

29 Jul 2022

PONE-D-21-19478

Power Posture Effects on Approach and Avoidance Decisions in Response to Social Threat

PLOS ONE

Dear Dr. Metzler,

Thank you for submitting your manuscript to PLOS ONE.  I apologize again for the considerable delay in reviewing process due to recurring significant difficulties in securing reviewers for your submission.  After careful consideration and feedback from two experts in the field, we feel that it has merit but does not fully meet PLOS ONE’s publication criteria as it currently stands. Therefore, we invite you to submit a revised version of the manuscript that carefully addresses the points raised during the review process.

If you choose to submit a revision, please make sure to respond point-by-point to Reviewers' comments found below, paying particular attention to the technical and statistical aspects of your work (including declaring where applicable, whether or not the data were normally distributed, two- or one-tailed tests used and why, and whether the tests were corrected for multiplicity).   

Please submit your revised manuscript within six months from this date as afterwards, any revision has to be considered a new submission.  If you will need more time than this to complete your revisions, please reply to this message or contact the journal office at plosone@plos.org.  Please include the following items when submitting your revised manuscript:

We look forward to receiving your revised manuscript.

Thank you for considering PLOS ONE for reporting your research.

Kind regards,

Sasha

Alexander N. 'Sasha' Sokolov, Ph.D.

Academic Editor

PLOS ONE

Comments from Staff Editor: We note that one or more reviewers has recommended that you cite specific previously published works. As always, we recommend that you please review and evaluate the requested works to determine whether they are relevant and should be cited. It is not a requirement to cite these works. We appreciate your attention to this request.

Journal Requirements:

5. We note that Figure 1 includes an image of a participant in the study. 

6. Thank you for submitting the above manuscript to PLOS ONE. During our internal evaluation of the manuscript, we found significant text overlap between your submission and the following previously published work, of which you are an author.

- https://tel.archives-ouvertes.fr/tel-02372963/file/these_METZLER_Hannah_2018.pdf

Please revise the manuscript to rephrase the duplicated text, cite your sources, and provide details as to how the current manuscript advances on previous work. Please note that further consideration is dependent on the submission of a manuscript that addresses these concerns about the overlap in text with published work.

Reviewers' comments:

Reviewer's Responses to Questions

**Comments to the Author**

1. Is the manuscript technically sound, and do the data support the conclusions?

Reviewer #1: Partly

Reviewer #2: Yes

2. Has the statistical analysis been performed appropriately and rigorously? 

Reviewer #1: Yes

Reviewer #2: Yes

3. Have the authors made all data underlying the findings in their manuscript fully available?

Reviewer #1: Yes

Reviewer #2: Yes

4. Is the manuscript presented in an intelligible fashion and written in standard English?

Reviewer #1: Yes

Reviewer #2: Yes

5. Review Comments to the Author

Reviewer #1: The authors report one experiment in which participants took first a neutral and then a power-related posture (expansive or contracted). After these manipulations, individuals were administered a decision task in which they decided on a computer screen to move their mouse cursor to a chair near or far from an angry or fearful looking individual. This decision was taken as an index for “I want to sit near or far from the individual”. Results revealed as expected a change in behavioral decisions due to posture: Angry individuals were even more avoided when participants had taken contracted postures compared to neutral postures. There was no change for expansive postures nor for trials with fearful individuals.

The study is well written and the results are interesting. However, given the heated debate concerning power posing effects, I was surprised that the authors conducted only a single study. As they state themselves, the observed effect was smaller than expected, and thus the study slightly underpowered. Furthermore, results were only achieved for anger in the contracted posture condition. One might thus say that hypotheses were only partially fulfilled. Moreover, some aspects of the theoretical reasoning are not entirely clear to me: The authors argue that effects of power posing might be found only for social behavior. For what reasons? Typically, researchers study power-related behavior, feelings etc. as consequences based on an embodiment explanation (i.e., the posture induces/reinstantiates a feeling of power). Many effects can also be explained via activation of semantic concepts (i.e., the semantic concept of power is activated, so that associated content is also more accessible). It is true that power postures also (maybe mainly) have a social message, but what is the assumed underlying mechanism of the effect? Embodiment? The authors also mention that none of the participants could identify fear and anger as the displayed emotions. Thus, on which processes should the effects be based? On nonconscious ones? In sum, I can unfortunately not recommend publication of the study as it is. With further data, however, the study would be a very interesting one.

Major points:

1. Theoretical points: The authors report on a topic which has – as they state themselves – received considerable debate, replication attempts, meta-analyses and reviews in the last years. Given the central importance of this debate for their research, I suggest that the authors are a bit more detailed and explicit about the findings that emerged throughout this debate and the link to their current study. Namely, that effects on other outcome measures beyond feeling were mainly not replicated (Jonas et al., 2017). The authors explicate that most behaviors (for which the effect could not be replicated) would have been non-social and that research regarding social behavior would have been scarce until now. What is lacking, however, is an explanation why power posing should influence social behavior, but not non-social (but power-related behavior, such as, e.g., risk taking).

With regard to their research and the specific emotions of anger and fear, the authors explain on p.6 why power posing should influence actions towards an angry or fearful other. Some of the mentioned research takes an appraisal approach, that is, this research takes as a starting point that the feeling of anger goes along with the appraisal of being in control/high power, or vice versa: if one feels in control, it is more likely to develop anger when confronted with an opponent, rather than fear. With regard to this issue, it is interesting to consider their effects. Do they think shared appraisals are the reasons for the observed effects? (see the review of Angie, Connelly, Waples, & Kligyte, 2011, for effects of specific emotions on judgment and decision-making). I would like to have the assumed underlying process or mechanism more outlined.

2. Gender of participants and stimuli: The authors explain that the decision of taking only men is empirically motivated. Is there also a theoretical explanation? Perhaps, men have a stronger power motive, or power posing is something which typically men use to demonstrate their power but not women? Given this choice, is the effect the same for male and female stimulus persons? Perhaps, the effect only plays a role for male stimuli? Or varies quantitatively depending on stimulus gender?

3. Task instructions and processing of facial expressions: Participants were instructed to keep their eyes on the fixation cross during the task. Perhaps as a consequence, the authors state on p.11 that “none [of the participants] could specifically identify anger and fear as the two displayed emotions”. Thus, on which processes did participants base their decisions to “approach” or “avoid” one person if it was not the perceived emotional facial expressions? Do the authors assume that the responsible processes are nonconscious ones? They must assume that anger and fear are nonconsciously processed and influence the decision. It is, however, debated whether specific emotions are nonconsciously processed on a specific level (see, e.g., Rohr, Degner, & Wentura, 2015).

4. The authors report that taking the postures took two minutes and each experimental block on average 6.5 minutes. I wondered how long the posture effect holds. Perhaps results are limited to the first block of experimental trials after the posture only?

Minor points:

• The authors term their task “approach-avoidance task”. However, participants’ task is not directly related to approaching the other. It might be said that a decision for the chair farer away is one of avoiding the other, but speaking of approach is not entirely correct. Maybe this is a point that should be discussed in order not to mingle the results with other approach-avoidance tasks.

• On p.7 136ff the authors outline shortly results of previous studies. They speak of the increased avoidance of angry faces. In comparison to what?

• On p.9 they describe how the total number of trials is made with regard to the conditions. This is very difficult to understand. Could you please rephrase/outline a bit more?

Reviewer #2: The authors studied the effects of power posing on approach and avoidance decisions. Participants were assigned to a high power pose or a low power pose group and engaged in a virtual stool-selection task while seeing angry or fearful persons sitting on stools. After low power posing, participants avoided fearful individuals whereas high power posing had no effect. The study is interesting and authors studied body positions in relation to a hierarchy-relevant topic: approach/avoidance behavior. Research on body positions is relevant and can progress this controversial field. However, before recommending publication, I think the authors should address the following points:

1) The theory section is not up-to-date and several hierarchy terms are mixed. I think the authors should provide more clarity about which effects of body positions are valid and which are not valid (see Körner et al., 2022). For example, the authors repeatedly cite the meta-analysis by Gronau et al. (2017). This meta-analysis relies on a very small sample size and only analyzes the effect of power posing on feelings of power. Thus, the generalizability of this meta-analysis is very limited.

Further, the field suffered from consensus and different kinds of body positions have been mixed despite different meanings of different degrees of expansiveness. Therefore, it would be helpful to clarify what body positions were actually studied: Poses (in line with power posing research) representing dominance (see Körner & Schütz, 2020; Witkower et al., 2020).

This leads to the next point: The authors quote findings from studies on power, dominance, status, etc. This is fine. However, it should be more clearly stated which effect pertains to which hierarchy variable (power is not dominance, see Blader & Chen, 2014). Much of the rationale relies on findings about dominance (e.g., almost the entire second paragraph about animals and body size in the Introduction). This is completely fine as recent research suggests that power poses are actually dominance poses (Körner & Schütz, 2020; Witkower et al., 2020), but again, more clarity should be provided. For example, the authors can use the term pose instead of posture to join consensus-building in this field. The theoretical rationale can more heavily describe what are correlates of dominance and how can dominant or submissive nonverbal displays affect approach/avoidance behavior. It is fine to still refer to power, as dominance often leads to power (in terms of influence and control), but it should be clearly stated what hierarchy construct is relevant in which sentence (e.g., writing dominance status makes it unclear whether the authors speak about dominance or about social status).

Blader, S. L., & Chen, Y.-R. (2012). Differentiating the effects of status and power: A justice perspective. Journal of Personality and Social Psychology, 102(5), 994–1014.

Körner, R., Röseler, L., Schütz, A., & Bushman, B. J. (2022). Dominance and prestige: Meta-analytic review of experimentally induced body position effects on behavioral, self-report, and physiological dependent variables. Psychological Bulletin. Advance online publication.

Körner, R., & Schütz, A. (2020). Dominance or prestige: A review of the effects of power poses and other body postures. Social and Personality Psychology Compass, 14(8), Article e12559.

Witkower, Z., Tracy, J. L., Cheng, J. T., & Henrich, J. (2020). Two signals of social rank: Prestige and dominance are associated with distinct nonverbal displays. Journal of Personality and Social Psychology, 118(1), 89–129.

2) The authors wrote “Yet, feelings as well as many other investigated behaviours were assessed using explicit self-reports, which are susceptible to demand effects” (p. 4). Behaviors are typically not assessed with self-reports because self-reports tap into feelings, thoughts, and self-evaluations. The sentence thus needs revision.

Moreover, the authors wrote “cognitively complex social behaviour, such as sales negotiation, cheating or planning to take revenge or to volunteer (Cesario & Johnson, 2017; Strelan et al., 2014; Yap et al., 2013).” Strelan et al. measured revenge with a self-report instrument. Thus, their DV represents rather a self-evaluation than a behavior. The sentence thus needs revision.

3) It may help to backup the sentence on p. 5, lines 96-98, with a reference.

4) p. 8, lines 459-165: This section should be removed. The most recent meta-analytical evidence shows that men and women do not differ in their effects of body positions (see Körner et al., 2022). The same applies to the last paragraph of the discussion (p. 22, lines 521-526) – it should be removed. Instead, it should be clearly stated that it is a limitation to study only men. Yet, as the results of other body position studies do not differ between men and women the authors may speculate that the same effects may be found with women.

5) The power/sensitivity analyses are very detailed. It is great that the authors provide so much elaboration on that part, however, it distracts somewhat. I recommend removing the first sensitivity analysis (pp. 8-9) because in the results section another, more important sensitivity analysis follows. Perhaps, the second sensitivity analysis could be included in a footnote to increase the flow of the manuscript. Moreover, the effect of body positions on self-reports and behaviors is small to medium and thus I would refrain from describing the study as high-powered. It may help to describe the effect sizes used in the power/sensitivity analyses in words so that readers instantly understand that the participant number provides enough power to detect large effects but not enough power to detect small effects.

6) It is great that approach/avoidance responses as an elementary behavior was studied as DV. However, I see two limitations: First, the virtual task is described as preventing demand effects. Yet, the within-subjects design makes demand effects more plausible (Greenwald, 1976). Thus, it may be better to frame this point more carefully (i.e., avoid to describe your study as almost free from any demand effects). Second, the stool-choosing scenario might be criticized as lacking ecological validity. Indeed, a behavior was studied but it is unknown whether the findings also apply to real-world settings.

Greenwald, A. G. (1976). Within-subjects designs: To use or not to use? Psychological Bulletin, 83(2), 314–320.

7) In the paragraph on p. 19 (lines 437-450), the references are messed up. The Weick et al. (2017)-reference does not refer to body positions and thus does not fit with the corresponding sentence. The Elkjaer et al. (2020)-reference was heavily criticized. Thus, both references should be removed. The year of Guinotes work is missing and the names of the 2003-reference (very likely Keltner et al.,) are missing.

Related to this: The authors refer to the work by Nielsen (2017) as evidence of a non-significant effect of body positions. Nielsen used trait measures and the non-significant results are thus not surprising. It would be better to cite papers published in renowned journals instead.

8) Probably, most readers of this article will be psychologists. Therefore, it will be good to stick with APA style with respect to presenting statistics.

9) p. 21, l. 493: The authors wrote their study has relatively high statistical power. This is somewhat exaggerated as the power of the sensitivity analysis did not reach .80 and I would refrain from describing power as high as long as it is <.90.

10) In the conclusion, the authors used the word “mediated”. I was somewhat surprised because no mediation analyses are reported. It might be helpful to use another term to avoid confusion.

11) I wondered whether neuroticism may explain the results of the angry condition. Dominance as well as sense of power are positively related to emotional stability whereas submissiveness shows a negative relation. Thus, participants in the low power posing condition might show less emotional stability and thus avoid intense emotional, threatening situations. Other explanations might also be possible for the results. The paper may benefit from a deeper discussion of the main result.

Overall, the authors conducted an interesting study. However, I think the theory and discussion parts need intense revision. I wish the authors all the best with their research.

Robert Körner

6. PLOS authors have the option to publish the peer review history of their article (what does this mean?). If published, this will include your full peer review and any attached files.

Reviewer #1: No

Reviewer #2: No

---

## [Author Response · Author response to Decision Letter 0]

9 Jan 2023

Reviewers' comments:

Reviewer #1:

The authors report one experiment in which participants took first a neutral and then a power-related posture (expansive or contracted). After these manipulations, individuals were administered a decision task in which they decided on a computer screen to move their mouse cursor to a chair near or far from an angry or fearful looking individual. This decision was taken as an index for “I want to sit near or far from the individual”. Results revealed as expected a change in behavioral decisions due to posture: Angry individuals were even more avoided when participants had taken contracted postures compared to neutral postures. There was no change for expansive postures nor for trials with fearful individuals.

The study is well written and the results are interesting. However, given the heated debate concerning power posing effects, I was surprised that the authors conducted only a single study. As they state themselves, the observed effect was smaller than expected, and thus the study slightly underpowered. Furthermore, results were only achieved for anger in the contracted posture condition. One might thus say that hypotheses were only partially fulfilled. 

Moreover, some aspects of the theoretical reasoning are not entirely clear to me: The authors argue that effects of power posing might be found only for social behavior. For what reasons? Typically, researchers study power-related behavior, feelings etc. as consequences based on an embodiment explanation (i.e., the posture induces/reinstantiates a feeling of power). Many effects can also be explained via activation of semantic concepts (i.e., the semantic concept of power is activated, so that associated content is also more accessible). It is true that power postures also (maybe mainly) have a social message, but what is the assumed underlying mechanism of the effect? Embodiment? The authors also mention that none of the participants could identify fear and anger as the displayed emotions. Thus, on which processes should the effects be based? On nonconscious ones? In sum, I can unfortunately not recommend publication of the study as it is. With further data, however, the study would be a very interesting one.

We thank the reviewer for their careful feedback, and found all points they raised important. First, we can relate to the surprise that we conducted only a single study. The context explains why: The study was conducted in 2016, when most replications of power posing studies were not yet published, and as the last study during a PhD project. At the time, most studies had used only self-reports of behaviors (subject to demand effects), and used small samples with between-subject designs. The current study improved on these studies in many ways, with a substantially larger sample, by including a baseline measure without posture, a power analysis and higher power than most studies at the time, with transparent reporting, sharing data, code and materials, and by focusing on social behavior rather than less theoretically related non-social behaviors (e.g. economic risk taking). Testing 88 participants with our experimental design took 2h per individual, and was at the limit of our feasibility constraints. Nonetheless, this was an exploratory and novel study, with its hypotheses being tested for the first time. Only part of the hypotheses was confirmed with statistical significance, but the other half was formulated as speculative from the beginning, and still went in the assumed direction. We reported all of this very transparently. When methods are robust and reporting is transparent, partially confirmed hypotheses and slightly lower effect sizes than expected are not reasons to reject a paper. 

We agree that further data would make the study very interesting. Unfortunately, we cannot collect any further data for the following reasons: The data for the current manuscript was collected in 2016, as the last one during a PhD project. Challenges with conducting a PhD project on a very debated and rapidly evolving research field on power poses, as well as very long delays (above a full year at PlosOne) during the review of the manuscript have led to it only being under revision at PlosOne now. The scientific field has evolved considerably since 2016, and the first author of this manuscript has moved on to other academic positions. We have therefore reported the level of evidence the study contributes in its current form as accurately and transparently as possible, so that other researchers can decide whether to invest further resources on a replication. We are convinced that the study in its current form is a valuable contribution to the scientific literature. We now transparently mention all of this in the limitation section of the paper (p20, L501 (all line numbers refer to the version with accepted changes)): 

«Finally, it is important to transparently report on the level of evidence this study provides for power pose effects on low-level social behavior. This requires mentioning its exploratory nature, and the debate about replicability especially for early research in the field (up to 2017). We believe it provides clear preliminary evidence thanks to its many strength (robust experimental and statistical methods, a baseline measure, open data, code and materials, strong theoretical foundation on dominance poses as a social behavior relevant in threatening contexts). Yet, we are convinced that the medium effect sizes, given our sample size, would require replication before strong conclusions can be drawn. »

Major points:

1. Theoretical points: The authors report on a topic which has – as they state themselves – received considerable debate, replication attempts, meta-analyses and reviews in the last years. Given the central importance of this debate for their research, I suggest that the authors are a bit more detailed and explicit about the findings that emerged throughout this debate and the link to their current study. Namely, that effects on other outcome measures beyond feeling were mainly not replicated (Jonas et al., 2017).

We now address the replication debate explicitly and in detail early on in the introduction: p3, L46: “Carney, Cuddy and Yap [17] first tested whether adopting power poses for two minutes could significantly affect subjective feelings of power and control, risk-taking behavior and levels of testosterone and cortisol. Although they found confirmatory evidence, later studies could not reproduce these results [18–26]. An intense debate about whether any of the later published power pose effects were reliable followed. Two recent meta-analyses summarizing all studies fulfilling certain criteria provide evidence for a statistical difference between expansive and contracted poses for self-reported feelings and overt behavioral outcomes, but not for hormones, after correcting for publication bias (n=96 reports [15], n=48 [27]). Both highlighted a lack of studies that allow conclusions about whether expansive or contracted power poses drive the effects. Preliminary findings based on only a few studies hint that only contracted poses may induce effects [27]. The most reliable available evidence to date, from large pre-registered studies and registered reports, suggests that the power pose effect on self-reported feelings of power is real, but small (n = 1002 participants [28], n=1071 [29]) Still, it seems likely that the effect arises partly from demand effects [15,29,30], and that publication bias remains an issue in the field [15,31].

The present study was conducted during the ongoing replication debate (in 2016), and needs to be interpreted in this context. Yet, it addresses some of the common issues in power posing studies, by assessing changes from a baseline to power-related poses [15,27], and using behavioral outcomes and a plausible cover-story rather than explicit instructions and self-reports, which increase the likelihood of demand effects [29,30]. Most importantly, it focuses on “lower” level processes of social interactions, such as the ability to perceive and respond to others’ social signals. With one exception [32], earlier work using behavioral outcomes has so far neglected such low-level social behaviors, focusing either on cognitively complex (and sometimes only reported) behaviors [e.g., 20,33,34] or non-social behaviors [e.g., 35,36].”

The authors explicate that most behaviors (for which the effect could not be replicated) would have been non-social and that research regarding social behavior would have been scarce until now. What is lacking, however, is an explanation why power posing should influence social behavior, but not non-social (but power-related behavior, such as, e.g., risk taking).

We did not mean to assert that power posing should only influence social behavior, and not non-social ones. When starting to investigate pose effects, we were surprised that past research mainly focused on non-social or cognitively complex social behaviors, neglecting “lower” levels processes of social interactions, such as the ability to perceive and respond to others’ social signals. We hope this is now more clearly communicated in the revised introduction. 

With regard to their research and the specific emotions of anger and fear, the authors explain on p.6 why power posing should influence actions towards an angry or fearful other. Some of the mentioned research takes an appraisal approach, that is, this research takes as a starting point that the feeling of anger goes along with the appraisal of being in control/high power, or vice versa: if one feels in control, it is more likely to develop anger when confronted with an opponent, rather than fear. With regard to this issue, it is interesting to consider their effects. Do they think shared appraisals are the reasons for the observed effects? (see the review of Angie, Connelly, Waples, & Kligyte, 2011, for effects of specific emotions on judgment and decision-making). I would like to have the assumed underlying process or mechanism more outlined.

We thank the reviewer for this comment which helped us reformulating our assumptions about how power poses could influence action decisions in social contexts. We didn’t take the appraisal approach as a starting point, neither the shared appraisal as the reasons for our observed effects. We don’t have arguments (for instance physiological or neural recordings) to defend that in our approach-avoidance task, observers feel anger (or fear) when they perceive other individuals express anger (or fear). Therefore, and contrarily to Angie et al. (2011)‘s review that aimed at assessing how specific felt emotional states have different implications for judgment and decision-making, we aimed at studying how specific perceived emotional displays differently affect approach/avoidance decisions as a function of their social meaning (and not the feeling that they provoke in the observers). 

We now explain this in the introduction: 

P4, L70: “A recent study indicated that adopting power poses impacts the perceptual salience of emotional facial displays [37]. In everyday life, emotional facial expressions not only inform others about the affective states and potential behavioral intentions of the emitter [38] but also convey action demands to the perceiver [39]. Accordingly, perceiving emotional expressions has a direct influence on the observer’s behavior [40], prompting motivational orientations that prepare the organism for appropriate responses. The present study investigated whether adopting power poses would also influence action decisions in the presence of emotional facial displays.”

[…]

P5, L89: “Given that social power is fundamentally about defending access to resources needed for survival against conspecifics [43,44], we focused on threat-related facial expressions of anger and fear. Encountering conspecifics that express threat-related facial expressions represents a potential menace to an individual’s resources and/or survival. The individual’s power will crucially determine how they can respond to social threats [44,45], that is, their action opportunities. While anger and fear are both of negative valence, they convey different social meanings [46,47]. Facial expressions of anger, by enhancing cues of strength and dominance [48], are signals of aggressive intent [49], a clear threat to the observer [46], which in most contexts leads to avoidance. Fearful displays, in contrast, signal both the presence of a potential danger [50] and a need for affiliation [51,52], and are thus more ambiguous in terms of avoidance and approach behaviors. Contrasting these two emotional expressions therefore allowed determining whether power poses differently impacts decisions based on clear versus ambiguous and subtler social cues.“

[...]

P5, L102: “Previous studies using this task [41,53,54] observed that participants tend to avoid angry individuals more often than fearful ones, without having been informed about the emotions, and most often, without being able to explicitly report which emotions were presented during the task. Both angry and fearful faces elicited quicker reactions than neutral faces, hinting that the increased avoidance of angry as compared to fearful faces cannot be explained by more efficient processing. Mennella et al. [54] and Grèzes et al. [53] further revealed that angry, and to a lesser extent fearful displays, increased the expected value of the action leading to avoidance. In other words, sitting far from a threatening individual was a highly motivational (desirable) outcome per se, even in the context of a laboratory task. Consistent with the proposal that flexible decision processes can include both explicit and implicit forecasting of action outcomes [55], these results provide evidence for an emotion-based process of value attribution that can influence action (approach/avoidance) selection very rapidly, and possibly outside consciousness [56,57].

Research further suggests that, during the process of decision-making, the brain specifies and assigns value to the potential actions available in the world using relevant information from both external (for instance, threat displays) and internal (i.e., the motivational, emotional, and cognitive state) contexts of the decision-maker [58–60]. Actions possibilities are thus constrained by the individual’s action capabilities [61], which can be rapidly recalibrated as a function of external [e.g., 62] or internal factors [63,64]. We therefore hypothesized that power poses could impact the computation of available action possibilities and associated expected value according to the level of power they embody, without influencing the ability to detect anger or fear.”

We also discuss this in more detail in the discussion: 

P19, L442: “One possible interpretation of the power pose effects observed in the present study relates to action possibilities. The perceived available action opportunities and their assigned expected value are constrained by individual’s action capabilities [61] that are rapidly recalibrated as a function of external [e.g. 62] or internal factors (e.g., fatigue [63,64]). When facing social threats, power poses could impact the computation of available action possibilities and associated expected value according to the level of power they embody. Appraisal theories also assume that appraisals depend on the person-environment relationship, which can change over time and circumstances [91–93]. Within this framework, power poses could change the appraisals of the motivational relevance of others’ emotional expressions and one’s perceived ability to react to the situation. A contracted pose may make it more difficult to confront a threatening other, and therefore lead to increased avoidance. This account is consistent with the approach-motivation theory of power and related evidence, which implies that high power individuals have more freedom to express their personality and emotions, whereas low power individuals need to adapt their behavior to situational circumstances and suppress their emotions [44,94,95]. This could explain why individuals in a powerless state tend to avoid intense emotional and threatening confrontations. Other possible interpretations of increased avoidance of angry individuals could be related to a change in mood or effort while holding the contracted pose. Future studies could try to differentiate between these different possible mechanisms.”

2. Gender of participants and stimuli: The authors explain that the decision of taking only men is empirically motivated. Is there also a theoretical explanation? Perhaps, men have a stronger power motive, or power posing is something which typically men use to demonstrate their power but not women? Given this choice, is the effect the same for male and female stimulus persons? Perhaps, the effect only plays a role for male stimuli? Or varies quantitatively depending on stimulus gender?

There are both empirical and theoretical reasons for testing only men. Empirical reasons include that we still believed postures could have hormonal effects when we designed this study (data collection was done in 2016), and that controlling menstrual cycle variation is complicated when not even knowing whether an effect exists and is robust. (This study was part of a larger project that investigated hormones). Although there are differences in power-related behaviors, regarding postural feedback effects, the largest meta-analysis to date (Körner et al., 2022) does not provide evidence for any gender effects. Given that male faces are generally perceived as more dominant, we did however check our main result split by stimulus gender. We found no differences between male and female stimuli (see Figure below or in the formatted version of the response to reviewers). 

3. Task instructions and processing of facial expressions: Participants were instructed to keep their eyes on the fixation cross during the task. Perhaps as a consequence, the authors state on p.11 that “none [of the participants] could specifically identify anger and fear as the two displayed emotions”. Thus, on which processes did participants base their decisions to “approach” or “avoid” one person if it was not the perceived emotional facial expressions? Do the authors assume that the responsible processes are nonconscious ones? They must assume that anger and fear are nonconsciously processed and influence the decision. It is, however, debated whether specific emotions are nonconsciously processed on a specific level (see, e.g., Rohr, Degner, & Wentura, 2015).

Although participants were not able to verbally (explicit emotional response labels) report which 2 emotions were present in the scene, the emotional expressions were processed, as participants were quicker to initiate their actions and took longer to reach the chairs in response to both threat-related displays as compared to neutral ones. Furthermore, some participants mentioned one or two of the emotions correctly in a much larger collection of other not presented emotions. We however do not want to assert that the emotions are nonconsciously processed, as being incapable of verbally reporting the emotions presented on the screen does not equate nonconscious processing. 

Yet, using the same protocol (Mennella et. al 2020, Grèzes et al. 2021), we showed that the influence of threat-related expressions on participant’s approach/avoidance decisions is mediated by changing the expected value of each available action option (e.g. a change in evidence accumulation, thanks to drift-diffusion models). Mennella et al. (2020) further showed, using EEG, that the early stimulus-locked neural encoding of value difference peaked before selective attention was allocated to emotional displays. P6/L110: “Consistent with the proposal that flexible decision processes can include both explicit and implicit forecasting of action outcomes [55], these results provide evidence for an emotion-based process of value attribution that can influence action (approach/avoidance) selection very rapidly, and possibly outside consciousness [56,57].”

Regarding whether specific emotions are nonconsciously processed on a specific level (see, e.g., Rohr, Degner, & Wentura, 2015), our scene was presented for 1400ms, and the stimulus-locked neural encoding of value difference peaked around 200ms, while the neural marker of attention peaked around 300ms (Mennella et al. 2020). In Wentura and colleagues (2015, 2020)’s papers, perceivable or masked primes of very short duration were used (24ms to 40 ms). Under clearly visible prime (more closely related to our protocol), they show that misattribution can be emotion-specific, rather than be confined to a mere valence distinction. Moreover, even when reducing prime visibility (30 ms), anger-related emotional scenes were clearly differentiated from fear (see Rohr et al. 2020), therefore suggesting that emotional expressions could be nonconsciously processed on a specific level.

4. The authors report that taking the postures took two minutes and each experimental block on average 6.5 minutes. I wondered how long the posture effect holds. Perhaps results are limited to the first block of experimental trials after the posture only?

We were fully aware of this time challenge when designing the study, and aimed at the best compromise between a still comfortable duration of the posture, and an enduring effect. Testing if the effect is limited to the first block of trials is not possible, however, because the 2 emotional and neutral scenes were fully randomized within each block. This means that we cannot control the amount of fear/anger stimuli in each half of block, splitting them by first second half would bias the results.

Minor points:

• The authors term their task “approach-avoidance task”. However, participants’ task is not directly related to approaching the other. It might be said that a decision for the chair farer away is one of avoiding the other, but speaking of approach is not entirely correct. Maybe this is a point that should be discussed in order not to mingle the results with other approach-avoidance tasks.

We are not sure if we understand the reviewer’s comment fully. Still, we clarify why we label the task approach-avoidance below, and have added a corresponding section in the S1 Appendix. We also mention it more briefly in the introduction. The task allows investigating spontaneous action decisions between two competing targets for action (two empty chairs), in the presence of two task-irrelevant individuals, one neutral and another one displaying a negative facial expression (either fear or anger). While we agree that participants’ task is not to explicitly approach or avoid emotional individuals, the choice of the chair close to the individual displaying threat-related facial expressions allows to decrease the distance between the chosen chair and the emotional individual, that is, approaching this individual. Conversely, choosing the chair far from this emotional individual decreases the distance between participant’s chosen chair and the emotional individual, i.e., avoiding her/him. As in everyday life, approaching someone implies moving closer to them, rather than further away, in the task. 

In the introduction: P4, L80: “The present task differs from previous approach-avoidance paradigms [42], during which participants are instructed to perform one movement (e.g. push or pull), which either results in approaching or avoiding valenced stimuli. In contrast to such forced-choice tasks [41], here, participants freely choose between two action alternatives in a scene representing an everyday environment (a waiting room), without the confounding effect of instructions, arbitrary movements or response labels (see S1 Appendix for a detailed comparison to previous approach-avoidance tasks).“

At the start of the S1 Appendix: “The approach-avoidance task we used [1] allows investigating spontaneous action decisions between two competing targets for action (two empty chairs) in the presence of two task-irrelevant individuals, one neutral and another one displaying a negative facial expression (either fear or anger). This experimental approach differs from existing compatibility paradigms (e.g. the Approach-Avoidance Task, AAT [2,3]) that require subjects to pull or push a joystick to categorize either the valence or some other task-irrelevant feature of stimuli, such as faces’ gender [for an overview, see 4]. In contrast to our free choice paradigm, in the AAT, participants have to perform one explicitly instructed action, which either results in approaching positive/avoiding negative stimuli (compatible trials), or approaching negative/avoiding positive stimuli (incompatible trials). Reaction times are usually faster in compatible than in incompatible trials [2,3,5,6]. 

However, a number of factors were shown to affect how subjects respond to the same emotional stimulus, such as the subject’s self-representation in space [7], or the explicit label (“approach”/”avoidance") assigned to the response movement [8,9]. For example, the same movement (e.g. arm flexion) has been labeled as either approach (e.g. for retrieving something desired) or avoidance (e.g. for moving away from a spider) across different studies [4]. Such factors could explain why AAT paradigms yielded contradicting findings regarding responses to angry and fearful expressions [6,10–13]. In contrast to these previous paradigms, the task by Vilarem et al. [1] allows participants to freely choose among alternatives in a scene representing an everyday environment, without the constraint of instructions, arbitrary movements or response labels. As in everyday life, it allows more closely examining how different alternatives for action compete to determine spontaneous approach-avoidance responses to emotional displays.”

• On p.7 136ff the authors outline shortly results of previous studies. They speak of the increased avoidance of angry faces. In comparison to what?

As we make the anger vs. fear comparison frequently in the paper, we did not repeat it every time. We still understand that this can be misleading, and inserted “compared to fearful” in this sentence (L 106).

• On p.9 they describe how the total number of trials is made with regard to the conditions. This is very difficult to understand. Could you please rephrase/outline a bit more?

We rephrased the sentence and it now reads (p8, L180): “Each individual in a pair once displayed each of the 2 emotions at 4 different intensity levels, and displayed the same neutral expression 4 times. Each individual was further once display on each side in the image (left/right) with each of these expressions. With 10 pairs of two individuals, this resulted in a total of 480 trials per session: 10 pairs x 2 individuals x 2 sides x (2 emotions x 4 intensities + 4x1 neutral expression).”

Reviewer #2:

The authors studied the effects of power posing on approach and avoidance decisions. Participants were assigned to a high power pose or a low power pose group and engaged in a virtual stool-selection task while seeing angry or fearful persons sitting on stools. After low power posing, participants avoided fearful individuals whereas high power posing had no effect. The study is interesting and authors studied body positions in relation to a hierarchy-relevant topic: approach/avoidance behavior. Research on body positions is relevant and can progress this controversial field. However, before recommending publication, I think the authors should address the following points:

1) The theory section is not up-to-date and several hierarchy terms are mixed. I think the authors should provide more clarity about which effects of body positions are valid and which are not valid (see Körner et al., 2022). For example, the authors repeatedly cite the meta-analysis by Gronau et al. (2017). This meta-analysis relies on a very small sample size and only analyzes the effect of power posing on feelings of power. Thus, the generalizability of this meta-analysis is very limited.

Further, the field suffered from consensus and different kinds of body positions have been mixed despite different meanings of different degrees of expansiveness. Therefore, it would be helpful to clarify what body positions were actually studied: Poses (in line with power posing research) representing dominance (see Körner & Schütz, 2020; Witkower et al., 2020).

We thank the reviewer for his excellent review, and have followed all of their recommendations. We have updated the introduction, it was effectively out-of-date after a reviewing period of more than a year. We made the distinction between prestige and dominance related postures, as it relates to a concern about the power posing literature we share. We also now mention the sample size of Gronau, and set this in relation to the other meta-analyses comprising all studies in the field. Still, we also find it important to highlight other signals of reliability of a study, and Gronau et al., (2017) is comprised of only registered reports, the most reliable publication format in the psychology literature. We therefore think both meta-analyses deserve to be mentioned. 

This leads to the next point: The authors quote findings from studies on power, dominance, status, etc. This is fine. However, it should be more clearly stated which effect pertains to which hierarchy variable (power is not dominance, see Blader & Chen, 2014). Much of the rationale relies on findings about dominance (e.g., almost the entire second paragraph about animals and body size in the Introduction). This is completely fine as recent research suggests that power poses are actually dominance poses (Körner & Schütz, 2020; Witkower et al., 2020), but again, more clarity should be provided. For example, the authors can use the term pose instead of posture to join consensus-building in this field. The theoretical rationale can more heavily describe what are correlates of dominance and how can dominant or submissive nonverbal displays affect approach/avoidance behavior. It is fine to still refer to power, as dominance often leads to power (in terms of influence and control), but it should be clearly stated what hierarchy construct is relevant in which sentence (e.g., writing dominance status makes it unclear whether the authors speak about dominance or about social status).

Blader, S. L., & Chen, Y.-R. (2012). Differentiating the effects of status and power: A justice perspective. Journal of Personality and Social Psychology, 102(5), 994–1014.

Körner, R., Röseler, L., Schütz, A., & Bushman, B. J. (2022). Dominance and prestige: Meta-analytic review of experimentally induced body position effects on behavioral, self-report, and physiological dependent variables. Psychological Bulletin. Advance online publication.

Körner, R., & Schütz, A. (2020). Dominance or prestige: A review of the effects of power poses and other body postures. Social and Personality Psychology Compass, 14(8), Article e12559.

Witkower, Z., Tracy, J. L., Cheng, J. T., & Henrich, J. (2020). Two signals of social rank: Prestige and dominance are associated with distinct nonverbal displays. Journal of Personality and Social Psychology, 118(1), 89–129.

We thank the reviewer for this comment. We have clarified this early on in the introduction, as well as in the abstract, and throughout the paper. We only use the term pose when referring to power poses, and have included the above references. 

Extract from the introduction: L37 (all line numbers refer to the clean revised manuscript with accepted changes): “Humans use analogous non-verbal displays to express social power. Expansive body poses signal high power, dominance, prestige and success [10–13], whereas contracted poses convey low power, submission, and defeat. Dominance and prestige are two distinct ways of achieving social power [14]. While they are both related to body expansion, upright vs. slumped postures were suggested to be associated with prestige whereas expansive vs. contracted power poses would be associated to dominance [13,15]. Moreover, direct eye gaze, absence of smiling behaviors, and larger distances between hands and feet would more specifically communicate dominance [16].”

2) The authors wrote “Yet, feelings as well as many other investigated behaviours were assessed using explicit self-reports, which are susceptible to demand effects” (p. 4). Behaviors are typically not assessed with self-reports because self-reports tap into feelings, thoughts, and self-evaluations. The sentence thus needs revision.

This is exactly the point we were trying to make (many studies only measuring self-reports about behaviors, rather than actual behaviors), thank you for pointing out this was still not clear. The sentence is now embedded in a general situation of our study in the context of the replication debate, and now reads (L62): “Yet, it addresses some of the common issues in power posing studies, by assessing changes from a baseline to power-related poses [15,27], and using behavioral outcomes and a plausible cover-story rather than explicit instructions and self-reports, which increase the likelihood of demand effects [29,30].”

Moreover, the authors wrote “cognitively complex social behaviour, such as sales negotiation, cheating or planning to take revenge or to volunteer (Cesario & Johnson, 2017; Strelan et al., 2014; Yap et al., 2013).” Strelan et al. measured revenge with a self-report instrument. Thus, their DV represents rather a self-evaluation than a behavior. The sentence thus needs revision.

Thank you for pointing this out. The sentence about these studies now reads (L67): “With one exception [32], earlier work using behavioral outcomes has so far neglected such low-level social behaviors, focusing either on cognitively complex (and sometimes only reported) behaviors [e.g., 20,33,34] or non-social behaviors [e.g., 35,36].“

3) It may help to backup the sentence on p. 5, lines 96-98, with a reference.

We have rewritten this to clearly express what we mean, and provide several references (L89): “Given that social power is fundamentally about defending access to resources needed for survival against conspecifics [43,44], we focused on threat-related facial expressions of anger and fear. Encountering conspecifics that express threat-related facial expressions represents a potential menace to an individual’s resources and/or survival. The individual’s power will crucially determine how they can respond to social threats [44,45], that is, their action opportunities. ” 

4) p. 8, lines 159-165: This section should be removed. The most recent meta-analytical evidence shows that men and women do not differ in their effects of body positions (see Körner et al., 2022). The same applies to the last paragraph of the discussion (p. 22, lines 521-526) – it should be removed. Instead, it should be clearly stated that it is a limitation to study only men. Yet, as the results of other body position studies do not differ between men and women the authors may speculate that the same effects may be found with women.

We removed these paragraphs and instead cite the reviewer’s own recent-meta-analysis, and also include a transparent explanation for why the study only included men in the limitation section. 

P7, L153: “Given gender-differences in a study of ours with the same facial stimuli [69, Chapter 4] we decided to maximize sample size per group by including only male participants. A recent meta-analysis suggests that power-pose effects do not differ between genders [15]. ”

P21, L486: “Despite the meta-analytic finding that power pose effects do not depend on gender [15], the inclusion of only male participants is a clear limitation of our study, This decision was due to the study being part of a larger project on power poses [69], which began before it was clear that effects on cortisol and testosterone did not replicate. We initially suspected hormonal mechanisms to underlie the poses’ effects on social behavior, and biological differences in the production of testosterone and the menstrual cycle require separate hormonal analyses per gender [102]. Therefore, most studies in the project focused only on men, whereas those that included women observed gender effects in face perception [69, Chapter 4]. Wanting to achieve robust results and maximize sample size per group in the context of the replication debate, within our feasibility constraints, we therefore included only men. Based on similar effects in both genders in Körner et al.’s [15] meta-analysis, it seems possible that our findings generalize to women. ”

5) The power/sensitivity analyses are very detailed. It is great that the authors provide so much elaboration on that part, however, it distracts somewhat. I recommend removing the first sensitivity analysis (pp. 8-9) because in the results section another, more important sensitivity analysis follows. Perhaps, the second sensitivity analysis could be included in a footnote to increase the flow of the manuscript. Moreover, the effect of body positions on self-reports and behaviors is small to medium and thus I would refrain from describing the study as high-powered. It may help to describe the effect sizes used in the power/sensitivity analyses in words so that readers instantly understand that the participant number provides enough power to detect large effects but not enough power to detect small effects.

Thank you for these recommendations. We have removed the sensitivity analysis part in the methods, and refrain from calling the study high-powered. Unfortunately, footnotes are not permitted in PlosOne. We have moved the sensitivity analysis results part, together with the part from the methods, to the S1 Appendix instead. 

6) It is great that approach/avoidance responses as an elementary behavior was studied as DV. However, I see two limitations: First, the virtual task is described as preventing demand effects. Yet, the within-subjects design makes demand effects more plausible (Greenwald, 1976). Thus, it may be better to frame this point more carefully (i.e., avoid to describe your study as almost free from any demand effects). Second, the stool-choosing scenario might be criticized as lacking ecological validity. Indeed, a behavior was studied but it is unknown whether the findings also apply to real-world settings.

Greenwald, A. G. (1976). Within-subjects designs: To use or not to use? Psychological Bulletin, 83(2), 314–320.

We agree with phrasing this more cautiously. In the discussion we now say, for instance, that (L377) “such elementary behaviors in response to task-irrelevant social stimuli may be less susceptible to experimental demand effects than explicit self-reports ”. We also mention this in the limitation section L486: “However, within-subject designs may also make demand effects more likely [15]. Since we measured actual behavior rather than self-reports, and only 2 participants suspected a link between the pose manipulation and the approach-avoidance task even after extensive debriefing (see S1 Appendix), demand effects are less likely to explain our results. ”

However, we disagree about the scenario lacking ecological validity. In contrast to other approach and avoidance tasks (forced choice pushing/pulling a joy stick to approach faces floating on a screen), as well as to behaviors measured in most other power posing studies (e.g. self-reports about intended behaviors like revenge in pen on paper questionnaires, or economic gambling (choosing to roll a dice), it resembles situations encountered in everyday life quite closely. Self-reports of feelings in a computer questionnaire (e.g. of power, self-esteem, control, etc) are also less likely to apply to real world settings than the approach-avoidance task used in the current study. We did therefore not change comments on ecological validity. 

7) In the paragraph on p. 19 (lines 437-450), the references are messed up.

The references did indeed get messed up in that paragraph, thank you for noticing this. 

The Weick et al. (2017)-reference does not refer to body positions and thus does not fit with the corresponding sentence.

The Weick et al. (2017) was mistakenly provided for a sentence that described Elkjaer et al. (2020). We have therefore removed Weick et al. (2017) from this sentence, and describe the study correctly elsewhere (L436-438).

The Elkjaer et al. (2020)-reference was heavily criticized. Thus, both references should be removed. The year of Guinotes work is missing and the names of the 2003-reference (very likely Keltner et al.,) are missing.

Thank you for drawing our attention to the problems with the meta-analysis by Elkjaer et al (2020), and now refer to the valuable critical commentary by the reviewer and colleagues (Körner, Röseler & Schulz, 2022). Yet, we found no other critiques of Elkjaer et al. Although we fully agree with all arguments in the critical commentary, we do not think that this commentary alone qualifies as “heavy criticism”. We further think that one of its arguments (lacking open science practices) could be used to disqualify many of the studies included in the meta-analysis (Körner et al., 2022), which also did not share code, data, materials, were not pre-registered, or lacked power. Based on the reviewer’s own commentary, the issues with Elkjaer et al. (2020) do not invalidate the statement we cite this meta-analysis for (preliminary evidence for only contracted postures having effects). To apply fair criteria to judge which studies to include, we have therefore decided not to entirely remove it, but to mention the problems associated with it, and emphasize the preliminary nature of the evidence. 

e.g. L55: “Preliminary findings based on only a few studies hint that only contracted poses may induce effects [27]. “

L395: “The fact that expansive poses did not significantly decrease avoidance of angry individuals, in contrast to our predictions, is consistent with preliminary evidence from a recent meta-analysis of the power posing literature, which hints that only contracted poses may change behaviour in comparison to neutral ones [27]. Due to issues associated with Elkjær et al.’s meta-analysis [16], and the small number of studies including neutral conditions, we can only speculate if this applies in general, or only applies in specific studies and contexts. “

Related to this: The authors refer to the work by Nielsen (2017) as evidence of a non-significant effect of body positions. Nielsen used trait measures and the non-significant results are thus not surprising. It would be better to cite papers published in renowned journals instead.

After examination, we find it hard to judge if the journal is problematic, since it is an open-access, peer-reviewed and double-blind journal published by the Canadian Center of Science and Education. Still, we agree that measuring optimism, self-esteem and problem-solving confidence are not the most likely to be affected by power poses (like many other measures used in other studies), and have removed the reference.

8) Probably, most readers of this article will be psychologists. Therefore, it will be good to stick with APA style with respect to presenting statistics.

We are not entirely sure what the reviewer refers to. Two guesses: 

1) We noticed we reported some p-values and effect sizes that cannot exceed 1 with a 0 before the decimal (0.05 etc). We have deleted these zeros. We also now report confidence intervals in square brackets [], and only report 95% for the CI when it has not been mentioned just before (according to the APA manual).

2) In case the reviewer is referring to LMEMs as the statistical method we use, we chose them instead of ANOVA’s because ANOVA’s are not appropriate for datasets with multiple trials and subjects (DeBruine & Barr, 2021, https://doi.org/10.1177/2515245920965119). We still chose to provide ANOVAs in addition because most readers will be psychologists, as you say. Psychological scientists are rapidly adopting LMEMs (Meteyard & Davies, 2020, 10.1016/j.jml.2020.104092), but the APA style guide does not yet provide specific reporting guidelines. We follow the reporting recommendations from Meteyard & Davis (2020). As a solution for this second point, we now mention the ANOVA analysis very early on in our results section, so readers can refer to it if they prefer. L13, L301: “(see S1 Appendix for results and effect sizes in the form of ANOVAs and Cohen’s d for central comparisons)”

9) p. 21, l. 493: The authors wrote their study has relatively high statistical power. This is somewhat exaggerated as the power of the sensitivity analysis did not reach .80 and I would refrain from describing power as high as long as it is <.90.

Given how this field of research has evolved since 2016 (when this study had relatively high power compared to other studies), we have deleted this comment. Instead, we refer to “reasonable statistical power” throughout. We want to emphasize that studies rarely are as transparent about the power they reach as we are when we include a post-hoc sensitivity analysis. Our power-calculation in advance was 80%, which the majority of power posture studies does not reach (this is what we referred to with the word relative in “relatively high statistical power”). We do not want the evidence provided in this study to be down-valued because of our transparency. Therefore, we chose to explain the following in the section on sensitivity analysis in the Appendix: “Although LMEMs generally have higher statistical power, we chose to calculate power via ANOVA, because power calculation for LMEMs requires simulations (DeBruine & Barr, 2021). Our sensitivity analysis thus likely underestimates the true statistical power of our analysis.”

10) In the conclusion, the authors used the word “mediated”. I was somewhat surprised because no mediation analyses are reported. It might be helpful to use another term to avoid confusion.

We now use the words “accompanied” instead. In the conclusion section (L421, p19): 

“The observed influence on action decisions in response to anger was not accompanied by differences in anger processing, but ...”

11) I wondered whether neuroticism may explain the results of the angry condition. Dominance as well as sense of power are positively related to emotional stability whereas submissiveness shows a negative relation. Thus, participants in the low power posing condition might show less emotional stability and thus avoid intense emotional, threatening situations. Other explanations might also be possible for the results. The paper may benefit from a deeper discussion of the main result.

We have added a much deeper discussion of the main result. Regarding neuroticism, we could not find any good references demonstrating the link between emotional stability/neuroticism and current dominance/power status. Given that neuroticism is a personality trait, we also wonder if adopting poses is actually likely to affect it. Instead, the approach-motivation theory of power predicts that high power individuals can express their personality (including neuroticism) and emotions more freely, while low power individuals need to adapt their behavior to situational circumstances and inhibit their emotions (Keltner et al., 2003). This could explain why individuals in a powerless state, including those with higher neuroticism, avoid intense emotional and threatening situations. We refer to this possibility in the discussion. 

“One possible interpretation of the power pose effects observed in the present study relates to action possibilities. The perceived available action opportunities and their assigned expected value are constrained by individual’s action capabilities [60] that are rapidly recalibrated as a function of external [e.g. 61] or internal factors (e.g., fatigue [62,63]). When facing social threats, power poses could impact the computation of available action possibilities and associated expected value according to the level of power they embody. Appraisal theories also assume that appraisals depend on the person-environment relationship, which can change over time and circumstances [90–92]. Within this framework, power poses could change the appraisals of the motivational relevance of others’ emotional expressions and one’s perceived ability to react to the situation. A contracted pose may make it more difficult to confront a threatening other, and therefore lead to increased avoidance. This account is consistent with the approach-motivation theory of power and related evidence, which implies that high power individuals have more freedom to express their personality and emotions, whereas low power individuals need to adapt their behavior to situational circumstances and suppress their emotions [43,93,94]. This could explain why individuals in a powerless state tend to avoid intense emotional and threatening confrontations. Other possible interpretations of increased avoidance of angry individuals could be related to a change in mood or effort while holding the contracted pose. Future studies could try to differentiate between these different possible mechanisms. ”

Overall, the authors conducted an interesting study. However, I think the theory and discussion parts need intense revision. I wish the authors all the best with their research.

Robert Körner

Thank you for all these valuable and constructive comments. They have really helped us to greatly improve and update our introduction and discussion. Your feedback, coming from an expert who is up to date with the power posing field was very valuable for us. We are looking forward to future interactions with you!

---

## [Decision Letter · Decision Letter 1]

19 Apr 2023

PONE-D-21-19478R1Power pose effects on approach and avoidance decisions in response to social threatPLOS ONE

Dear Dr. Metzler,

thank you for submitting your revised manuscript to PLOS ONE.  I have now heard back from the Reviewers and second their view that the manuscript has been significantly improved.  After careful consideration, however, we would like to convey the remaining concerns and ask you to carefully address them, especially those by Reviewer 2, in another revision of your submission.   Please submit your revised manuscript with point-by-point response to the points raised within six months from this date as beyond that point, any revision has to be considered a new submission.  If you will need more time than this to complete your revisions, please reply to this message or contact the journal office at plosone@plos.org. Please include the following items when submitting your revised manuscript:A rebuttal letter that responds to each point raised by the academic editor and reviewer(s). You should upload this letter as a separate file labeled 'Response to Reviewers'.A marked-up copy of your manuscript that highlights changes made to the original version. You should upload this as a separate file labeled 'Revised Manuscript with Track Changes'.An unmarked version of your revised paper without tracked changes. You should upload this as a separate file labeled 'Manuscript'.If applicable, we recommend that you deposit your laboratory protocols in protocols.io to enhance the reproducibility of your results. Protocols.io assigns your protocol its own identifier (DOI) so that it can be cited independently in the future. For instructions see: https://journals.plos.org/plosone/s/submission-guidelines#loc-laboratory-protocols. Additionally, PLOS ONE offers an option for publishing peer-reviewed Lab Protocol articles, which describe protocols hosted on protocols.io. Read more information on sharing protocols at https://plos.org/protocols?utm_medium=editorial-email&utm_source=authorletters&utm_campaign=protocols.

We look forward to receiving your revised manuscript.

Kind regards,

Sasha

Alexander N. (Sasha) Sokolov, Ph.D.

Academic Editor

PLOS ONE

Journal Requirements:

Reviewers' comments:

Reviewer's Responses to Questions

**Comments to the Author**

1. If the authors have adequately addressed your comments raised in a previous round of review and you feel that this manuscript is now acceptable for publication, you may indicate that here to bypass the “Comments to the Author” section, enter your conflict of interest statement in the “Confidential to Editor” section, and submit your "Accept" recommendation.

Reviewer #1: All comments have been addressed

Reviewer #2: (No Response)

2. Is the manuscript technically sound, and do the data support the conclusions?

Reviewer #1: Yes

Reviewer #2: Yes

3. Has the statistical analysis been performed appropriately and rigorously? 

Reviewer #1: Yes

Reviewer #2: Yes

4. Have the authors made all data underlying the findings in their manuscript fully available?

Reviewer #1: Yes

Reviewer #2: Yes

5. Is the manuscript presented in an intelligible fashion and written in standard English?

Reviewer #1: Yes

Reviewer #2: No

6. Review Comments to the Author

Reviewer #1: The authors took great care in responding to the raised critical points. I really like the present version of the manuscript. The authors now report the different aspects of power-posing research, the problems of replicability and the gap their research fills in much more detail. They are also more transparent with regard to the exploratory aspects of their study and the slight underpowering of their research. Importantly, they also provide more background about the possibly underlying processes which allows readers to compare it to other power-posing research (i.e., perhaps power-posing effects are based on different mechanisms depending on the paradigm). I find it thus very important that the authors outline the differences between their approach-avoidance task and other paradigms. Their action decision is one a more conceptual level (i.e., you have to know that sitting on that chair reduces/increases distance) while a constant body posture is taken, whereas in other’s the arm movement is the DV.

As a small point: I think there is no need to expand that much on Rohr et al. 2015 and related articles. :- ) (I think you can delete or shorten to one or two sentences). I really was just curious on how participants processed the expressions. So, it seems that it was attentional unawareness (i.e., like in the inattentional blindness Gorilla movie) for most participants.

Reviewer #2: Thank you for inviting me to re-review the manuscript titled, “Power pose effects on approach and avoidance decisions in response to social threat”. I read the response letter and the revised manuscript. Overall, I feel that the authors have addressed my concerns almost completely. However, I still have a few minor points:

1) In response to a large point by Reviewer 1, the authors wrote, “The present study was conducted during the ongoing replication debate (in 2016), and needs to be interpreted in this context.“ This sounds somewhat like, “The study is not methodological up-to-date but it was several years ago and this is why it should be published because it fulfills publishing criteria some years ago”. I think this is problematic because the study needs now – in 2023 – to convince the scientific community. The authors should remove the sentence and refer to limitations in the Discussion. Perhaps it might also help to name the study a “pilot study” or “pilot experiment”.

2) I am not convinced with the authors’ response to my comment 7. If the authors still decide to keep the Elkjaer et al. (2022) meta-analysis in the manuscript they should accompany the reference with the criticism (https://doi.org/10.1177/1745691620984474).

Further, it seemed that there is a misunderstanding in reading the commentary: Not the lack of open science practices in empirical power posing studies is criticized but the lack of open science practices in the Elkjaer et al. (2022) meta-analysis is criticized. The authors did not share their data and analysis code (also not on request), which is why we cannot fully trust such a recent paper in this controversial field, in which preregistration and open science practices are very important to regain trust. Moreover, as the Elkjaer et al. (2022) meta-analysis includes not even 50% of the studies analyzed in the more recent meta-analysis (Körner et al., 2022), it seems more than questionable why contracted body positions might have an effect.

Thus, I strongly recommend to (1) refer to the meta-analysis only together with the critical commentary because otherwise readers might be misled, and (2) do not claim at any place in the manuscript that other meta-analyses suggest the effect in power posing studies might be driven by the contractive body position group (we still don’t know this but your own study might add to the open question).

3) I am not a native English speaker but I feel that the manuscript needs to be proofread.

Signed, Robert Körner

7. PLOS authors have the option to publish the peer review history of their article (what does this mean?). If published, this will include your full peer review and any attached files.

Reviewer #1: **Yes: **Dr. Michaela Rohr

Reviewer #2: No

---

## [Author Response · Author response to Decision Letter 1]

4 May 2023

Dear Editor,

Thank you for giving us the opportunity to revise our manuscript “Power pose effects on approach and avoidance decision in response to social threat” (Ms. Ref. No.: PONE-D-21-19478R1) by Emma Vilarem, Adrian Petschen, Julie Grèzes, and myself, Hannah Metzler. 

We also thank the Reviewers for their comments which helped improve our article once more. We have responded to all points raised by the journal and reviewers. 

Please find attached a detailed, point-by-point response to the Reviewers. We provide a marked-up and an unmarked version of our manuscript with all changes they requested. Line and page numbers refer to the marked-up version. We think that the manuscript has further improved and hope that you will consider it suitable for publication in PLOS ONE.

We thank you very much for your time and consideration of our manuscript and look forward to hearing from you. 

Sincerely,

Hannah Metzler and Julie Grèzes

Review Comments to the Author

A note regarding editor and reviewers:

Reference numbers and page numbers below refer to the marked-up manuscript with tracked changes. Reference numbers in the final unmarked manuscript differ, because marked-up version still contains the references that were cut.

Reviewer #1: 

The authors took great care in responding to the raised critical points. I really like the present version of the manuscript. The authors now report the different aspects of power-posing research, the problems of replicability and the gap their research fills in much more detail. They are also more transparent with regard to the exploratory aspects of their study and the slight underpowering of their research. Importantly, they also provide more background about the possibly underlying processes which allows readers to compare it to other power-posing research (i.e., perhaps power-posing effects are based on different mechanisms depending on the paradigm). I find it thus very important that the authors outline the differences between their approach-avoidance task and other paradigms. Their action decision is one a more conceptual level (i.e., you have to know that sitting on that chair reduces/increases distance) while a constant body posture is taken, whereas in other’s the arm movement is the DV.

As a small point: I think there is no need to expand that much on Rohr et al. 2015 and related articles. :- ) (I think you can delete or shorten to one or two sentences). I really was just curious on how participants processed the expressions. So, it seems that it was attentional unawareness (i.e., like in the inattentional blindness Gorilla movie) for most participants.

Dear Dr. Rohr, 

Thank you very much for reviewing our manuscript a second time, and for your constructive and positive feedback. Regarding the last small point: we shortened the description of how participants processed the expressions in the manuscript, and have inserted this part of the manuscript below. We also thank you for suggesting the term attentional unawareness, which we may use in the future to describe our findings.

Best, 

Hannah Metzler

The shortened section now reads (page 6, tracked changes are visible in the manuscript and the PDF version of this letter): 

Mennella et al. [54] and Grèzes et al. [53] further revealed that angry and to a lesser extent fearful displays, increased the expected value of the action leading to avoidance. In other words, sitting far from a threatening individual was a highly motivational (desirable) outcome per se, even in the context of a laboratory task. 

Research further suggests that action possibilities are constrained by the individual’s action capabilities [61], which can be rapidly recalibrated as a function of external [e.g., 62] or internal factors [63,64]. 

Reviewer #2:

A note regarding editor and reviewers: 

Reference numbers and page numbers below refer to the marked-up manuscript with tracked changes. Reference numbers in the final unmarked manuscript differ, because marked-up version still contains the references that were cut.

Thank you for inviting me to re-review the manuscript titled, “Power pose effects on approach and avoidance decisions in response to social threat”. I read the response letter and the revised manuscript. Overall, I feel that the authors have addressed my concerns almost completely. 

Dear Dr. Körner, 

Thank you for reviewing our manuscript a second time, your comments helped improve our manuscript once more. 

However, I still have a few minor points:

1) In response to a large point by Reviewer 1, the authors wrote, “The present study was conducted during the ongoing replication debate (in 2016), and needs to be interpreted in this context.” This sounds somewhat like, “The study is not methodological up-to-date but it was several years ago and this is why it should be published because it fulfills publishing criteria some years ago”. I think this is problematic because the study needs now – in 2023 – to convince the scientific community. The authors should remove the sentence and refer to limitations in the Discussion. Perhaps it might also help to name the study a “pilot study” or “pilot experiment”.

We have removed the sentence, and rephrased the limitation section in the discussion. This involved moving the sentences marked in blue from lines 498-503 to the limitations section (line 517-525), and instead cutting and restructuring some other sentences. 

The limitation section now reads (page 22, line 516): “Despite these strengths, this study is not without limitations. First, within-subject designs were suggested to increase the risk of demand effects [15]. In the past, the majority of studies on postural feedback effects used explicit self-reports about behaviors [97–100], which appear to be susceptible to demand effects. Indeed, the pose effect on explicit feelings of power was found to be larger in subjects familiar with “power posing”, in a meta-analysis of six pre-registered and highly-powered studies [30], as well as in a similarly large pre-registered study [29]. In the present study, however, only 2 participants suspected a link between the pose manipulation and the approach-avoidance task even after extensive debriefing (see S1 Appendix), indicating that the influence of poses on action decisions in response to task-irrelevant stimuli may reduce such demand effects. 

Second, we only included male participants in the present study. This decision was due to the study being part of a larger project on power poses [70], which began before it was clear that effects on cortisol and testosterone did not replicate. We initially suspected hormonal mechanisms to underlie the poses’ effects on social behavior, and biological differences in the production of testosterone and the menstrual cycle require separate hormonal analyses by gender [104]. Therefore, most studies in the project focused only on men, whereas those that included women observed gender effects in face perception [70, Chapter 4]. Wanting to achieve robust results and maximize sample size per group in the context of the replication debate, within our feasibility constraints, we therefore included only men. As Körner et al.’s [15] meta-analysis revealed that power pose effects do not depend on gender, it seems possible that our findings may generalize to women.” 

2) I am not convinced with the authors’ response to my comment 7. If the authors still decide to keep the Elkjaer et al. (2022) meta-analysis in the manuscript they should accompany the reference with the criticism (https://doi.org/10.1177/1745691620984474).

Further, it seemed that there is a misunderstanding in reading the commentary: Not the lack of open science practices in empirical power posing studies is criticized but the lack of open science practices in the Elkjaer et al. (2022) meta-analysis is criticized. The authors did not share their data and analysis code (also not on request), which is why we cannot fully trust such a recent paper in this controversial field, in which preregistration and open science practices are very important to regain trust. Moreover, as the Elkjaer et al. (2022) meta-analysis includes not even 50% of the studies analyzed in the more recent meta-analysis (Körner et al., 2022), it seems more than questionable why contracted body positions might have an effect.

Thus, I strongly recommend to (1) refer to the meta-analysis only together with the critical commentary because otherwise readers might be misled, and (2) do not claim at any place in the manuscript that other meta-analyses suggest the effect in power posing studies might be driven by the contractive body position group (we still don’t know this but your own study might add to the open question).

We are sorry that our response was not clear. We fully agree that open science practices are important to regain trust in research on power poses, and understood that this was about Elkjaer et al. (2020) itself. Regarding your recommendations, in the current revision, we made sure that we accompany the reference to Elkjaer et al. (2020) with your critical commentary (https://doi.org/10.1177/1745691620984474), and have removed one mention of Elkjaer et al. (2020) in the introduction. Our earlier revision cited your meta-analysis instead of the commentary, this was an unintentional mistake for which we apologize. We also now use the year of its online publication (2020), to make clear that your commentary that mentions the year 2020 in its title refers to this publication. However, we think that not mentioning Elkjaer et al. (2020)’s findings at all is unjustified, notably in the discussion. We find the inclusion criteria applied in Elkjaer et al., (2020) reasonable for the specified purpose of aggregating only effects of body-feedback, only effects of whole body manipulations, and only behavioral or affective outcomes. The meta-analysis thus focuses on a smaller, but consistent set of studies, which are quite similar to our current study, and therefore important to mention once.

Here are the sections we edited (tracked changes are visible in the marked-up manuscript, and the PDF version of this response letter): 

 • Introduction (page 3): “Two recent meta-analyses provide evidence for a statistical difference between expansive and contracted poses for self-reported feelings and overt behavioral outcomes, but not for hormones, after correcting for publication bias (n=96 reports [15], n=48 reports [27, but see 28]). Both highlighted a lack of studies that allow definitive conclusions about whether expansive or contracted power poses drive the effects. ” 

 • Discussion (page 19, line 417): “Expansive poses did not significantly decrease avoidance of angry individuals, in contrast to our predictions. While our result appears to be consistent with a recent meta-analysis suggesting that only contracted poses may change behavior in comparison to neutral poses [27], the issues raised about this meta-analysis [28], and the small number of studies including neutral conditions prevent any definitive conclusion at this point.” 

3) I am not a native English speaker but I feel that the manuscript needs to be proofread.

Signed, Robert Körner

We have proofread the manuscript, and run a grammar correction and language improvement tool over it. All language improvement changes are marked in the manuscript with tracked changes.

---

## [Decision Letter · Decision Letter 2]

26 May 2023

Power pose effects on approach and avoidance decisions in response to social threat

PONE-D-21-19478R2

Dear Dr. Metzler,

thank you for submitting your revised manuscipt and efforts while improving your work.  We’re pleased to inform you that your manuscript has been judged scientifically suitable for publication and will be formally accepted for publication once it meets all outstanding technical requirements.

Kind regards and best wishes,

Sasha

Alexander N. 'Sasha' Sokolov, Ph.D.

Academic Editor

PLOS ONE

Additional Editor Comments (optional):

Reviewers' comments:

Reviewer's Responses to Questions

**Comments to the Author**

1. If the authors have adequately addressed your comments raised in a previous round of review and you feel that this manuscript is now acceptable for publication, you may indicate that here to bypass the “Comments to the Author” section, enter your conflict of interest statement in the “Confidential to Editor” section, and submit your "Accept" recommendation.

Reviewer #2: All comments have been addressed

2. Is the manuscript technically sound, and do the data support the conclusions?

Reviewer #2: Yes

3. Has the statistical analysis been performed appropriately and rigorously? 

Reviewer #2: Yes

4. Have the authors made all data underlying the findings in their manuscript fully available?

Reviewer #2: Yes

5. Is the manuscript presented in an intelligible fashion and written in standard English?

Reviewer #2: Yes

6. Review Comments to the Author

Reviewer #2: (No Response)

7. PLOS authors have the option to publish the peer review history of their article (what does this mean?). If published, this will include your full peer review and any attached files.

Reviewer #2: No

---

## [Editor Report · Acceptance letter]

10 Aug 2023

PONE-D-21-19478R2 

Power pose effects on approach and avoidance decisions in response to social threat 

Dear Dr. Metzler:

I'm pleased to inform you that your manuscript has been deemed suitable for publication in PLOS ONE. Congratulations! Your manuscript is now with our production department. 

Kind regards, 

on behalf of

Dr. Alexander N. Sokolov 

Academic Editor

PLOS ONE